# Update on the GOSAT TANSO–FTS SWIR Level 2 retrieval algorithm

Yu Someya[1], Yukio Yoshida[1], Hirofumi Ohyama[1], Shohei Nomura[1], Akihide Kamei[1], Isamu Morino[1], Hitoshi Mukai[2], Tsuneo Matsunaga[1], Joshua L. Laughner[3], Voltaire A. Velazco[4, 5], Benedikt Herkommer[6], Yao Té[7], Mahesh Kumar Sha[8], Rigel Kivi[9], Minqiang Zhou[10], Young Suk Oh[11], Nicholas M. Deutscher[5], and David W.T. Griffith[5]

[1]Earth System Division, National Institute for Environmental Studies, Tsukuba, Japan
[2]Center for Climate Change Adaptation, National Institute for Environmental Studies, Tsukuba, Japan
[3]Jet Propulsion Laboratory, California Institute of Technology, Pasadena, CA, USA
[4]Deutscher Wetterdienst, Meteorological Observatory Hohenpeissenberg, 82383, Germany
[5]Centre for Atmospheric Chemistry, School of Earth, Atmospheric and Life Sciences, University of Wollongong, NSW 2522, Australia
[6]Institute of Meteorology and Climate Research, Karlsruhe Institute of Technology, Germany
[7]Laboratoire d'Études du Rayonnement et de la Matière en Astrophysique et Atmosphères (LERMA-IPSL), Sorbonne Université, CNRS, Observatoire de Paris, PSL Université, 75005 Paris, France
[8]Royal Belgian Institute for Space Agency (BIRA-IASB), Brussels, Belgium
[9]Space and Earth Observation Centre, Finnish Meteorological Institute, Tähteläntie 62, 99600 Sodankylä, Finland
[10]CNRC & LAGEO, Institute of Atmospheric Physics, Chinese Academy of Sciences, Beijing, China
[11]National Institute of Meteorological Sciences, 33, Seohbuk-ro, Seogwipo-si, Jeju-do, 63568, Republic of KOREA

*Correspondence to*: Yu Someya (someya.yu@nies.go.jp)

**Abstract.** The National Institute for Environmental Studies has provided the column-averaged dry-air mole fraction of carbon dioxide and methane ($XCO_2$ and $XCH_4$) products (L2 products) obtained from the Greenhouse gases Observing SATellite for more than a decade. Recently, we updated the retrieval algorithm used to produce the new L2 product, V03.00. The main changes from the previous version (V02) of the retrieval algorithm are the treatment of cirrus clouds, the degradation model of the Thermal And Near-infrared Spectrometer for carbon Observation – Fourier Transform Spectrometer (TANSO-FTS), solar irradiance spectra, and gas absorption coefficient tables. The retrieval results from the updated algorithm showed improvements in fitting accuracies in the $O_2$ A, weak $CO_2$, and $CH_4$ bands of TANSO-FTS, although the residuals increase in the strong $CO_2$ band over the ocean. The direct comparison of the new product obtained from the updated (V03) algorithm with the previous version V02.90/91 and the validations using the Total Carbon Column Observing Network revealed that the V03 algorithm increases the amount of data without diminishing the data qualities of $XCO_2$ and $XCH_4$ over land. Further, the negative bias of $XCO_2$ is larger than that of the previous version over the ocean, and bias correction is still necessary. Additionally, the V03 algorithm resolves the underestimation of the $XCO_2$ growth rate compared with the in situ measurements over the ocean recently found using V02.90/91 and V02.95/96.

## 1 Introduction

The Greenhouse gases Observing SATellite (GOSAT) is the joint project of the Japan Aerospace Exploration Agency, the Ministry of the Environment, and the National Institute for Environmental Studies (NIES), and is the first satellite dedicated to monitoring greenhouse gases (GHGs), such as carbon dioxide ($CO_2$) and methane ($CH_4$) from space (Yokota et al., 2009). Since its launch on January 23, 2009, it has constantly provided the global concentrations of GHGs for more than 13 years. Additionally, the successor of GOSAT, GOSAT-2 was launched in 2018 and is also still in orbit. The sensor onboard GOSAT, Thermal And Near-infrared Sensor for carbon Observation (TANSO) consists of two instruments, the Fourier Transform Spectrometer (FTS; Kuze et al., 2009) and the Cloud and Aerosol Imager (CAI). TANSO-FTS measures the spectral regions ranging as 0.758–0.775 μm (12,900–13,200 $cm^{-1}$), 1.56–1.72 μm (5,800–6,400 $cm^{-1}$), and 1.92–2.08 μm (4,800–5,200 $cm^{-1}$) in short-wavelength infrared (SWIR) and 5.56–14.3 μm (700–1,800 $cm^{-1}$) in the thermal infrared (TIR) regions, with a spectral interval of approximately 0.2 $cm^{-1}$ and a spectral resolution (defined as the full width at half maximum of the instrumental line shape function) of 0.262 – 0.367 $cm^{-1}$ in the SWIR bands (Kuze et al., 2009). The trace gas concentrations or cloud properties have been estimated from the SWIR bands (Yoshida et al., 2011; 2013) and the TIR band (Saitoh et al., 2009; 2016; Ohyama et al., 2012; Someya et al., 2016; 2020).

The SWIR bands measure the reflected sunlight to estimate column-averaged dry-air mole fractions of $CO_2$ ($XCO_2$) and $CH_4$ ($XCH_4$). NIES provides the SWIR Level 2 (L2) product, which contains $XCO_2$ and $XCH_4$ retrieved using the GOSAT SWIR spectra (Yoshida et al., 2011; Yoshida et al., 2013). The L2 product is used to estimate the global surface fluxes of $CO_2$ and $CH_4$ and the resulting concentration distributions provided as Level 4 products (Maksyutov et al., 2013). Other groups have developed retrieval algorithms for GOSAT and provided column-averaged dry-air mole fractions of $CO_2$ and $CH_4$ (Butz et al., 2011; Parker et al., 2011; Oshchepkov et al., 2011; O'Dell et al., 2012; Cogan et al., 2012; Kikuchi et al., 2016; Noël et al., 2021; Taylor et al., 2022). The major differences among these algorithms include, e.g., the treatments of atmospheric particles or radiative transfer calculations. The algorithms are roughly classified into two categories considering whether multiple scattering by clouds and aerosols which are critical sources of error is explicitly considered, or not.

The current version of the NIES SWIR L2 product is the version 02 series (V02.xx), which has been improved from the previous version in several ways such as the treatment of the aerosols (Yoshida et al., 2013). Owing to this improvement, both the biases and precisions against the ground-based measurements, the Total Carbon Column Observing Network (TCCON; Wunch et al., 2011), are much less than 1% for $XCO_2$ and $XCH_4$. However, there are still some issues to address. First, the systematic structures in the spectral residuals still exist in the retrieval results. Second, the increase of data amount in the L2 product is further required. In addition, inconsistencies in the annual $CO_2$ growth rate compared with the in situ measurements were recently found in the V02.90/91 and V02.95/96 products. Therefore, the retrieval algorithm was updated to V03 to address these issues. Herein, we present an algorithm for the new version of the NIES SWIR L2 product, V03.xx.

## 2 Current retrieval algorithm

### 2.1 NIES V02 retrieval algorithm

The retrieval algorithm for the SWIR L2 product developed at NIES (Yoshida et al., 2013) is a full physics-based algorithm, that explicitly considers the scattering processes by particles in the atmosphere in the radiative transfer calculation. Four spectral ranges, 12,950–13,200 cm$^{-1}$ (O$_2$ A sub-band), 6,180–6,380 cm$^{-1}$ (WCO$_2$ sub-band), 5,900–6,150 cm$^{-1}$ (CH$_4$ sub-band), and 4,800–4,900 cm$^{-1}$ (SCO$_2$ sub-band) are used for the retrievals. The retrieval algorithm is based on the maximum a posteriori solution (Rodgers, 2000). This method obtains a solution to the state by minimizing the cost function,

$$J(x) = [y - F(x, b)]^T S_\epsilon^{-1}[y - F(x, b)] + (x - x_a)^T S_a^{-1}(x - x_a)$$

where $y$ represents the measurement vector, $F$ denotes a forward model, $x$ is a state vector, $b$ denotes a model parameter vector, $S_\epsilon$ represents a measurement error covariance matrix, $x_a$ denotes an a priori state vector, and $S_a$ represents an a priori covariance matrix. In the NIES retrieval algorithm, the state vector contains the profiles of the gases (CO$_2$, CH$_4$, and H$_2$O) and two types of aerosols, surface albedo over land, wind speed over the ocean surface, surface pressure (Ps), vertically constant temperature shift, zero level offset for the O$_2$ A sub-band, and wavenumber dispersions for each sub-band. The a priori values of CO$_2$ and CH$_4$ are obtained from the NIES transport model (NIES-TM; Saeki et al., 2013), and those of aerosol concentrations are from the Spectral Radiation-Transport Model for Aerosol Species (SPRINTARS; Takemura et al., 2000). Meteorological information is taken from the grid point value (GPV) objective analysis data using the global spectral model (GSM) provided by the Japanese Meteorological Agency. The atmosphere is divided into 15 vertical layers for radiative transfer calculations; the gas optical thickness is calculated every 12 sub-layers in each layer, i.e., 180 sub-layers in total.

### 2.2 Motivation for algorithm update

Although the number of TANSOFTS observations in the daytime is approximately 9000 per day, less than 10% of the total observations pass through the cloud screening and quality control filters to produce the L2 product. Thus, increasing the available number of observations for the L2 product is desirable to increase the TANSO-FTS measurement coverage. The existence of clouds is the main reason for the decrease in the available number of observations. The V02 algorithm discriminates the cloudy scenes using CAI images and the water vapor saturation band near 2 μm, which are mainly used to discriminate optically thick and cirrus clouds, respectively (Yoshida et al., 2011). In the V03 algorithm, cirrus cloud screening using the water vapor saturation band is not applied. Instead, we attempt to retrieve cirrus clouds simultaneously with the GHGs to increase the number of observations.

The spectral residuals obtained from the V02 retrievals have systematic wavenumber-dependent structures. The main causes of these structures are the uncertainties of the solar irradiance spectra and spectroscopic parameters of the trace gases. These datasets are updated to reduce the systematic residuals. In addition, the common use of these datasets with the

GOSAT-2 retrievals makes the L2 product from both satellites homogeneous. The homogeneousness of the products makes their continuous and simultaneous use easy.

According to the validation study, biases in the retrieval results of XCO$_2$ and XCH$_4$ without bias correction indicate spatial and temporal dependencies, significantly affecting the flux inversions and production of the Level 4 products. Therefore, NIES provides the bias-corrected product (V02.95/96) as well as the bias-uncorrected one (V02.90/91). Recently, we found that the growth rate of the XCO$_2$ estimated from the GOSAT L2 product, V02.95/96 or V02.90/91 over the ocean is lower than that over land, or the validation data such as TCCON and in situ measurements (NIES GOSAT project, 2021). Due to this issue, the GOSAT L2 V02.97/98 product with additional correction applied to its long-term trend based on the bias-corrected V02.95/96 product, has been released. The sensitivity degradation of TANSO–FTS could be the main cause of this issue. In this study, the degradation model is updated to decrease the temporal dependencies.

## 3 Updates on the retrieval algorithm

### 3.1 Treatment of cirrus clouds

The 2 μm band cloud screening mentioned in Section 2.2 is not performed in the V03 algorithm. Alternatively, the spectral band, 5150 to 5200 cm$^{-1}$ (H$_2$O sub-band) is additionally used in the retrieval to simultaneously estimate the cloud optical thickness (COT) and cloud top pressure (CTP) with GHG concentrations. We assume a single cloud layer with a pressure thickness of 30 hPa in which the ice particles with an effective dimension of 20 μm are homogeneously distributed. The optical property of the ice particle is obtained from the generalized habit mixture model proposed by Baum et al., (2011). The a priori values of COT and CTP are 0.1 and 150 hPa globally. If the retrieved COT is larger than 0.1, the post-screening process rejects the observation.

### 3.2 Degradation model

The radiometric sensitivity of TANSO-FTS has been degraded exponentially as a function of time relative to the pre-launch calibration with spectral dependencies. The V02 algorithm considers this degradation based on the degradation model developed by Yoshida et al. (2012). The V03 algorithm employs the model recently developed by Someya and Yoshida (2020). This model was constructed from the temporal variations of the principal components obtained from on-orbit solar irradiance calibration data using a diffuser plate to distinguish and separately evaluate the components. Although the new degradation model used in V03 and the previous one used in V02 are usually similar, the differences were found with several spectral dependencies. These differences increase with time because the new degradation model was constructed based on the longer data period. Therefore, the update of the degradation model is expected to affect the temporal dependencies of retrieval accuracy. The retrieval results obtained using this model show that the temporal dependency of the XCO$_2$ bias against the TCCON measurement is reduced with respect to those using the current model in Someya and Yoshida (2020).

### 3.3 Solar irradiance spectra

The solar irradiance spectra used in the V02 algorithm were created using the baseline estimated from the report by Dr. R. Kurucz and the Fraunhofer lines personally provided by Dr. G. C. Toon (Yoshida et al., 2013). The baseline and Fraunhofer lines were updated in V03. The baseline was estimated using the Total and Spectral Solar Irradiance Sensor–1 Hybrid Solar Reference Spectrum (TSIS–1 HSRS; Coddington et al., 2021). Fraunhofer lines were obtained from version 2016 of Toon (2015b).

### 3.4 Gas absorption coefficient database

In the radiative transfer calculation of retrieval processing, gas absorption coefficients are obtained by interpolating look-up tables (LUTs) as the functions of temperature, pressure, and wavenumber. The LUTs are created using several databases, and the referenced databases were updated (Table 1.) Mendonca et al. (2017) found that the $CH_4$ retrieval using HITRAN2008 depends on the solar zenith angle. In the V02 retrievals, the residuals at several $H_2O$ absorption lines increase with increasing water vapor because of the large uncertainties in spectroscopic parameters of $H_2O$. These problems can be resolved or mitigated by the updates. Associated with this update of LUTs, the scaling factor for $O_2$ absorption (see Section 2.3 of Yoshida et al., 2013 for details) is updated to 0.99556. Owing to the updates, the gas absorption coefficient database used in V03 retrievals is common to that used in the NIES SWIR L2 retrieval algorithm for TANSO–FTS–2 on GOSAT–2.

### 3.5 Other changes

In the NIES retrieval algorithm, the empirical noise model was estimated as the quadratic function of the signal-to-noise ratio to define the error covariance matrix (Yoshida et al., 2013). The coefficients of the functions in the V03 algorithm were updated due to the abovementioned changes. The empirical noise is not applied to the $H_2O$ sub-band.

Post-screening is applied to the result after the retrievals, and one of the screening items is the spectral residual. The retrieval results with the mean squared of the residuals normalized with spectral noise larger than the thresholds are screened and not included in the L2 product. The thresholds were re-evaluated as 1.2, 1.2, 1.2, and 1.3 for the $O_2$ A, $WCO_2$, $CH_4$, and $SCO_2$ sub-bands, respectively. The threshold is undefined for the $H_2O$ sub-band due to its large variability in water vapor concentrations.

Tables 2 and 3 summarize the retrieval setup for the V03 algorithm and the pre/post-screening procedures for the V02 and V03 algorithms.

## 4 Results

### 4.1 Spectral fitting accuracy

Figure 1 shows the averaged spectral residuals after the post-screening at each sub-band obtained in April 2009 and April 2020 over land from V02.90. The plots were normalized with the maximum radiances in each spectral range. These are differences between the simulated radiance spectra using posterior states and the observed spectra. In each sub-band presented in the figure, the residuals exhibit some spectral dependencies. In the $O_2$ A sub-band, the residuals at the edges of the sub-band are larger than those in the central region and the structures of the $O_2$ absorption are seen. In the $WCO_2$ and

$CH_4$ sub-bands, the residuals have relatively fine structures related to the gas absorptions, though those at the edges and in the center are flattened. Figure 2 shows the spectral residuals same as Fig. 1, except that V03.00 is used. Compared with Fig. 1, the wavenumber dependencies of the residuals are decreased and the retrievals seem to be well fitted in Fig. 2. Same figures over the ocean are shown in Fig. 3 and 4. Same as over land, the fitting accuracies of V03.00 are found to be better than those of V02.90 in the $O_2$ A, $WCO_2$, and $CH_4$ sub-bands. However, in the $SCO_2$ sub-band, the residual has a significant

spectral dependency, and it corresponds to the $CO_2$ absorption structure. The root mean squares of the averaged spectral residuals in April 2020 shown in the figures are summarized in Table 4. The values from V03.00 are lower than the values from V02.90/91 and the spectral fitting accuracies are improved except for the $SCO_2$ sub-band over the ocean.

The abovementioned differences in spectral residuals between V02.90 and V03.00 are mainly owing to the update of solar irradiance and gas absorption cross-section database. This is because the treatment of clouds has a smaller impact on the fine structure of the residuals, and there are slight spectral dependencies of differences between the new and old degradation

models shortly after the launch. The update of solar irradiance decreased the relatively large wavenumber dependencies, such as the large residuals around 6,375 $cm^{-1}$ and the large wavenumber dependency around 6,000 $cm^{-1}$ shown in Fig. 3. Updating the gas absorption cross-section database significantly improves the fitting accuracy in the $CH_4$ sub-band and substantially decreases the fine structure of the residuals. The $O_2$ A sub-band is flattened, and the differences between the center and edges of the sub-band are decreased by both the updates of solar irradiance and gas absorption coefficients.

However, in the $O_2$ A sub-band, some differences between 2009 and 2020 remain. One possible reason of this is the degradation model. The number of components of principal component analysis used to construct the degradation model in the $O_2$ A sub-band is smaller than the other band because the contributions of the primary components are large. The temporal differences are possibly due to the contributions by the other components which are not considered in the

construction of the degradation model.

Figure 4 shows the significant spectral dependencies of the residuals obtained from V03.00 in the $SCO_2$ sub-band over the ocean. In this figure, the baselines of the simulated radiance spectra seem to have some biases. This is introduced by the update of the solar irradiance spectra because there are some differences in the spectral baseline between the old and updated spectra particularly in Band 3 (see Appendix A and supplementals). Over the ocean, the surface state is described only by the

185 surface wind speed in the retrieval and the spectral baseline is not adjusted (unlike that in over land). The spectral structure

corresponding to $CO_2$ absorption is found in this figure. This can be result of changes in retrieved $CO_2$ to reduce residuals due to baseline bias. This can lead to a bias in the retrieved $XCO_2$. Therefore, we need to precisely evaluate the calibration data such as obtained in the Railroad valley campaign using the updated solar irradiance spectra to improve the fitting accuracy especially on the $SCO_2$ sub-band.

## 4.2 Global distribution of the retrieval results

In this section, we show the difference in the retrieval results between V02.90/91 and V03.00. The data from the launch to 2021 are used for both versions. Global distributions of the retrieved $XCO_2$, $XCH_4$, and the number of observations for V02.90/91, V03.00, and their differences are shown in Fig. 5 and Fig. 6 separately for over land and the ocean. The $XCO_2$ from V03.00 over land is approximately the same as that from V02.90/91. Conversely, over the ocean, the $XCO_2$ from V03.00 is 4.24 ppm lower than that from V02.90/91 for the match-up observations. This difference arises due to the spectral residual in the $SCO_2$ sub-band mentioned in Section 4.1.

The $XCH_4$ from V03.00 is lower than that from V02.90/91 globally. The changes in $XCH_4$ are commonly shown with a magnitude of approximately 8 ppb over land and the ocean. It is largely decreased in the middle and low latitudinal areas. Although it is difficult to isolate the impacts of each update on the retrieval results, our sensitivity test revealed that the $XCH_4$ over land changed by approximately 7 ppb depending on whether solar irradiance spectra are updated or not. On the other hand, the other test with the replacement of the gas absorption table shows smaller changes in $XCH_4$ over land (See appendix A). These may indicate that the decrease in $XCH_4$ is mainly because of the update of the solar irradiance spectra.

The temporal heatmaps of the differences in the monthly mean $XCO_2$ and $XCH_4$ between the versions are shown in Fig. 7. Differences in monthly mean $XCO_2$ get smaller with time, particularly over the ocean. This means the growth rate of $XCO_2$ from V03.00 is larger than that from V02.90/91. The long-term trend in $XCO_2$ is evaluated using the in situ measurements in section 4.4. Similar trends are also seen in $XCH_4$. Additionally, the seasonal variabilities of $XCH_4$ are larger than those of $XCO_2$, especially for the former period over the ocean. This is partly because the changes in $XCH_4$ over the ocean have latitudinal dependencies as shown in Fig. 6. The global distributions of the seasonal mean $XCO_2$ and $XCH_4$ in 2010 and 2021 are shown in Fig. 8. As seen in Fig. 5 – 7, the differences are smaller in the recent period, and they have the latitudinal dependencies. In addition, the latitudinal variations change seasonally as shown in Fig. 8. The increasing trend of $XCO_2$ in the high latitudes in Fig. 5 is introduced by the change in boreal spring (MAM) and this is not seen in boreal summer (JJA). A similar characteristic is also seen in $XCH_4$.

The number of observations over land is increased significantly because the 2 μm cloud screening is not applied in V03 retrievals. Because the $XCO_2$ values over land from V02.90/91 and V03.00 have only slight differences, the addition of the cirrus cloud parameters is effective to increase the number of observations. However, the number of observations over the ocean is decreased, except in the intertropical convergence zone where cirrus clouds frequently exist because the residuals in the $SCO_2$ sub-band are increased, and more observations are filtered through the post-screening process in the V03.00 retrieval. The numbers of observations from the V02.90/91 and V03.00 $XCO_2$ products are shown in Table 5. The V03.00

product increases the number of observations obtained over land and the mixed surface of land and ocean, by 12.7% and 22.3% compared with the V02.90/91 product, respectively. In opposite, it decreases by 20.3% over the ocean. Overall, the number of available observations from V03.00 is 2.3% larger than that from V02.90/91.

Figure 9 shows the global distributions of the ancillary parameters, the difference between the retrieved and a priori surface pressures ($\Delta Ps$), retrieved temperature shift, large-particle AOT, and the COT from V02.90/91 and V03.00. These results are obtained only from the observations that passed the post-screening process those with large AOT and COT (>0.1) are excluded. The general $\Delta Ps$ patterns are similar for V02.90/91 and V03.00. Over land, negative biases are slightly improved in V03.00. Over the ocean, positive biases are large in the high latitudes of the southern hemisphere for V02.90/91 and low latitudes for V03.00. The horizontal pattern of $\Delta Ps$ over land in the middle and low latitudes seems to correspond to that of the difference in $XCH_4$ shown in Fig. 5. Correlation coefficients between the changes in the retrieved surface pressure and those in $XCH_4$ from V02.90/91 and V03.00 are –0.57 over land and –0.64 over the ocean. The relatively large decrease in $XCH_4$ in low latitudes over the ocean could be partly attributed to the changes in $\Delta Ps$. For $XCO_2$, those are –0.57 over land and –0.11 over the ocean. Negative biases of temperature shift decreased globally for V03.00, and those over the ocean for V02.90/91 changed to slightly positive biases. Although the relatively large negative biases remain in inland China for V03.00, those in Europe and America for V02.90/91 become smaller for V03.00. The AOT of large particles at 1.6 μm decreased globally, especially over the ocean for V03.00. The COT is obtainable only for V03.00. Although the observations with large COT values are rejected by post-screening, the relatively large values are seen in the tropical regions, where cirrus clouds are frequently present.

Although the updated items do not independently affect the retrieval results and it is difficult to evaluate separately, the large causes of the change in the retrieved ancillary parameters are as follows from the sensitivity test retrievals (Appendix A). Temperature shift is increased globally by the update of the gas absorption coefficient tables. Surface pressure seems to be impacted by the replacement of solar irradiance because $\Delta Ps$ was changed by this update over land. The changes in surface pressure should contain two effects. One is the direct impact of the change in spectroscopy on the $O_2$ A sub-band. The other one is the impact through the change of $XCO_2$ introduced by the inconsistency of the spectral baseline in the $SCO_2$ sub-band. The behaviors of changes in AOT differ for land and the ocean. The changes in AOT are mainly affected by the addition of cirrus properties to the state vector over land. On the other hand, those over the ocean seem to be affected multiply by the updates. Figure 10 shows the time series of the ancillary parameters. V02.90/91 has a long-term temporal dependency on the retrieved surface pressure over land, temperature shift, AOT over the ocean. The pointing system of TANSO–FTS was switched from the primary system (PM–A) to the backup system (PM–B) on January 26, 2015. The trends differ for PM–A or PM–B, and they are larger in PM–A. For V03.00, those in surface pressure and AOT almost disappeared whereas that in temperature shift remains in PM–A.

## 4.3 Comparison with TCCON measurements

The retrieved $XCO_2$ and $XCH_4$ are validated using the TCCON measurements in this section. The TCCON sites used in this study are listed in Table B1. The GOSAT measurements used for the comparisons are selected within ±2° from each TCCON site. The TCCON measurements within ±30 min from the GOSAT measurement time are averaged for comparison. We used the data from the launch to 2021. Currently, the newest TCCON product, version GGG2020, is provided and we used this version in this analysis. However, not all sites have produced their full GGG2020 time series at the time of writing. The main changes between GGG2020 and the previous version, GGG2014, are found on the TCCON wiki page (https://tccon-wiki.caltech.edu/Main/DataDescriptionGGG2020)). The data amount of GGG2020 is currently smaller than that of GGG2014 because of stricter quality control processes, but much of these data should be recovered in the near future. In particular, measurements collected before 2011 are sparse.

The comparison results for V03.00 and V02.90/91 versus TCCON are shown in Table 6. Bias means the average of the differences between GOSAT and TCCON, and the standard deviations are calculated from these differences. The GOSAT measurements are categorized according to the surface state and the gain (high: H or middle: M) setting of the FTS measurement. The observations containing both the land and ocean surfaces in the instantaneous field of views of TANSO–FTS are not used here. The number of observations with gain H from V03.00 is larger than that from V02.90/91 over land. On the other hand, those with gain M from V03.00 are slightly smaller than those from V02.90/91. The sites used for gain M are only two sites, Pasadena and JPL which are very close to each other and located near Los Angels. Over the ocean, the number of observations from V03.00 decreases. There are no substantial changes in the standard deviations of the differences for $XCO_2$ and $XCH_4$ in all the situations (slightly worse in V03.00), although the biases are different between V03.00 and V02.90/91 in some cases.

The biases and standard deviations of the $XCO_2$ from V03.00 are close to those from V02.90/91 over land. Considering these results, the $XCO_2$ from V03.00 has similar qualities as that from V02.90/91 over land. Meanwhile, the bias of the $XCO_2$ from V3.00 is larger and more negative than that from V02.90/91 over the ocean. This issue is consistent with the results presented in Section 4.2 and is because of the fitting accuracy shown in Section 4.1. Therefore, the bias correction seems necessary for the $XCO_2$ from V03.00 over the ocean.

As shown in Section 4.2, the $XCH_4$ from V03.00 decreased from those from V02.90/91. Over land, the absolute values of the $XCH_4$ from V03.00 are slightly larger with gain H and significantly smaller with gain M than those from V02.90/91. Over the ocean, the bias from V03.00 is larger, although a smaller data amount is available. Therefore, we need to investigate the biases over the ocean with a larger amount of data in the future.

The validation results over land with gain H in the stricter match-up condition of ±0.1° are shown in Table 7 to investigate these differences more precisely. Because of the spatial variability of GHGs, the validation with the stricter condition is more reliable, especially for $XCH_4$. Unfortunately, there are no match-up data found over land with gain M and over the ocean in this match-up condition. In this table, the absolute values of bias and standard deviation of the $XCH_4$ from V03.00 are

smaller than those from V02.90/91. Therefore, the quality of the $XCH_4$ from V03.00 can be regarded as almost the same as or better than those from V02.90/91. Similar to the results from the looser match-up condition, the results of $XCO_2$ from V03.00 increase the number of observations and are slightly worse biases and standard deviations.

Inter-site and temporal variability of the differences between GOSAT and TCCON are investigated using the match-up condition of ±0.1°. The data with more than 10 match-up observations were used for both the investigations of inter-site and temporal variability. 10 TCCON sites (Burgos, Caltech, JPL02, Lauder02, Lauder03, Lamont, Paris, Saga, Sodankyla, and Tsukuba) were found as the match-up data sites for investigating inter-site variability Site biases, average site bias, and site-to-site variability were calculated as the mean differences from TCCON for individual sites, an average of site biases, and a standard deviation of site biases, respectively. The average site biases and the site-to-site variabilities from V03.00 are −0.01 and 1.74 ppm for $XCO_2$ and −2.14 and 9.33 ppb for $XCH_4$, respectively. Those from V02.90/91 are −0.02 and 1.72 ppm for $XCO_2$ and 5.99 and 9.12 ppb for $XCH_4$. The average site biases and the site-to-site variabilities of $XCO_2$ are similar for V03.00 and V02.90/91. For $XCH_4$, although the site-to-site variability from V03.00 is slightly higher than that from V02.90/91, the average site bias is smaller in V03.00. Temporal variability was calculated from the annual mean of the differences between GOSAT and TCCON. The time series of the annual mean differences are shown in Fig. 11. Temporal trends of the $XCO_2$ from V03.00 and V02.90/91 are similar after 2014 though the values from V02.90/91 are respectively large in 2012 and 2013. Although the values from V03.00 are generally lower than those from V02.90/91, the same trends are found for $XCH_4$. The standard deviations of the annual mean values from V03.00 and V02.90/91 are 0.42 and 0.52 ppm for $XCO_2$ and 1.44 and 2.06 ppb for $XCH_4$ respectively. Thus, V03.00 exhibits smaller temporal variability than V02.90/91 in this analysis. The decadal trends of the differences from V02.90/91 and V03.00 are –0.63±0.15 and –0.11±0.14 ppm/decade for $XCO_2$ and –2.41±0.84 and –0.37±0.77 ppb/decade for $XCH_4$. The consistencies of the decadal trend are slightly improved in V03.00.

### 4.4 Evaluating the long-term trend using in situ measurements

The TCCON sites used in the previous section were mainly obtained over land. However, as noted in Section 2.2, there is an issue with the decadal growth rate of $XCO_2$ estimated using the V02.90/91 product over the ocean. In this section, we evaluate the long-term trends of $XCO_2$ using in situ measurement data.

NIES has observed $CO_2$ via air sampling on ships (Tohjima et al., 2005), and at ground stations (Nomura et al., 2017; 2021) in southwestern Asia and the western Pacific Ocean for more than a decade. $CO_2$ in the upper troposphere has been observed by aircraft in the CONTRAIL project (Machida et al., 2008). In addition, NOAA Global Monitoring Laboratory has provided flask sampling and in situ measurement data on the western Pacific islands (Conway et al., 1994; Lan et al., 2022). The data used in this study are listed in Table B2. The products from these in situ measurements are appropriate to evaluate the GOSAT product in terms of the stability of data accuracy. Because these observations obtain the concentrations of the trace gases at the surfaces or at certain atmospheric levels, that are not column-averaged, they are not directly comparable with the $XCO_2$ obtained from GOSAT. Therefore, we only focus on the decadal increasing trend of $CO_2$ from both products

in this study. Further, we only evaluate the $CO_2$ trends because the comparison of $CH_4$ is more complicated due to its large vertical gradient and variability. For aircraft measurement, only the data obtained at altitudes of 5 km or higher were used. The 22 areas are defined using $12° \times 12°$ grid boxes and the $CO_2$ concentrations obtained from GOSAT and each in situ measurement platform were monthly averaged in each area for comparison. The locations of the in situ measurements and areas used in this analysis are depicted in Fig. 12. The monthly averaged values in each area from GOSAT and the in situ measurements are directly compared to investigate the difference in the decadal growth.

Figure 13 shows the time series of the differences between the $XCO_2$ from the GOSAT V02.90/91 or V03.00 product and $CO_2$ concentration from each in situ measurement platform. Here we used the data until 2020. The trend is estimated by the least-squared linear regression from the scatter data. Over land, the growth rates of $CO_2$ estimated from the GOSAT V02.90/91 and V03.00 products are consistent with that from the in situ measurements within 1 ppm/decade. This value is close to the difference between TCCON and the in situ measurements. On the other hand, the growth rate for V02.90/01 over the ocean is $1.68 \pm 0.14$ ppm/decade smaller than that from the in situ measurements. However, the difference in the growth rate for V03.00 is improved to $0.01 \pm 0.15$ ppm/decade although the biases are negatively large as shown in the previous sections. The differences in the growth rates between GOSAT and each platform are shown in Fig. 14. Over land, the absolute differences in the growth rates from V03.00 are smaller than those from V02.90/91 for ship and station measurements although that is slightly larger for aircraft. Over the ocean, the differences from V03.00 are smaller than those from V02.90/91 for all platforms. In particular, the large discrepancy of -2.7 ppm/decade with the station measurement in the V02.90/91 product was improved to -0.8 ppm/decade in the V03.00 product.

The main cause of this trend of the GOSAT V02.90/91 product over the ocean is estimated as the sensitivity degradation of TANSO–FTS. Although the degradation is considered in the V02 algorithm with the degradation model according to Yoshida et al. (2012), the degradations after 2012 are the expected ones. The error of this degradation model generates a gap in the spectral baseline between the observed and simulated spectra. The difference in trend is not significant over land because the gap can be adjusted by simultaneously retrieving surface albedo. In the NIES retrieval algorithm, only the wind speed is retrieved as a surface property over the ocean and not surface albedo. Therefore, the difference in the trend of $CO_2$ between GOSAT V02.90/91 and the in situ measurements could have resulted from the increasing error of the degradation model with time. This improvement of the trend of V03.00 over the ocean is mainly because of the update of the degradation model described in Section 3.2 as the other updates do not vary over time.

**4.5 Bias correction**

Because the V03.00 product has biases particularly for $XCO_2$ over the ocean, as shown in the previous sections, those should be corrected. We used TCCON GGG2014 for this bias correction because insignificant changes were found in $XCO_2$ between both versions and the available amount of data is larger than GGG2020. The site information of TCCON GGG2014 used in this study is listed in Table B3. The bias correction for $XCH_4$ is not processed here since those are largely changed between GGG2014 and GGG2020. Since the GGG2020 is not fully available, we plan to correct $XCH_4$ based on GGG2020

after more stations are published. The bias correction strategy is the same as that used in the V02.95/96 and V02.97/98 products (NIES GOSAT project, 2020). The bias correction of the $XCO_2$ for V03 is a function of AOT, $\Delta Ps$, and surface albedo at the $O_2$ A sub-band. Multiple linear regression analysis was used to estimate the coefficients. The TCCON data from 2009 to 2019 is used as the reference data. The changes in the $XCO_2$ from V03.00 after the correction are shown in Fig. 15. Over land, the corrections are generally positive although they are negative only in the high reflectance surface areas such as Sahara and Australia. The corrections over the ocean show similar positive values globally. The negative bias over the ocean revealed in the previous sections is corrected by this procedure. The mean changes and their standard deviations in $XCO_2$ by the bias correction are +0.55 ppm over land and +6.44 ppm over the ocean. The time series of the monthly mean changes by the bias correction is depicted in Fig. 16. The seasonal dependencies of the correction differ for the surface. Over land, the correction magnitude is large in boreal spring and summer. On the other hand, that is large in boreal winter over the ocean. This is because the ancillary parameters used in the bias correction are different by the surfaces and the common parameters also have different seasonal variations by the surfaces as shown in Fig. 10. The bias-corrected version of the $XCO_2$ product plans to be released as V03.05.

## 5. Summary and conclusions

The retrieval algorithm for the GOSAT TANSO–FTS SWIR L2 product from NIES was updated to generate the next version, the V03 product. The main changes in the V03 algorithm compared with the current retrieval algorithm (V02) are as follows:

1. COT and CTP are retrieved simultaneously with the GHGs instead of the cirrus cloud screening using the 2 μm band in the pre-screening processing
2. The degradation model of TANSO–FTS is updated to that of Someya and Yoshida (2020)
3. Solar irradiance spectra are updated to those produced from TSIS-1 HSRS and the version 2016 of Toon (2015b)
4. Gas absorption coefficient tables are updated using several new references

The retrieval results show that the spectral fitting accuracies are successfully improved, and the systematic spectral residuals in the V02.90/91 product are reduced in the $O_2$ A, $WCO_2$, and $CH_4$ sub-bands. Conversely, the residual in the $SCO_2$ sub-band increases over the ocean with a systematic spectral structure corresponding to the $CO_2$ absorptions. This increase in the residual is mainly attributed to spectral biases at baseline between observed and simulated spectra.

The amount of data from V03.00 is larger than that from V02.90/91 over land and the mixed surfaces mainly owing to the change in the treatment of clouds, although it is smaller over the ocean because of the residual in the $SCO_2$ sub-band. Overall, the amount of data from V03.00 increased by 2.3% compared with that from V02.90/91.

The direct comparison of V03.00 with V02.90/91 and the validation using TCCON measurements shows that the quality of $XCO_2$ from V03.00 is almost the same level as that from V02.90/91 over land—the update achieves an increase in the available data without reducing the quality of the retrieved $XCO_2$. On the other hand, the $XCO_2$ from V03.00 over the ocean is negatively biased and the bias correction is necessary. Although the bias $XCH_4$ over land with gain H from V03.00 is

slightly larger than that from V02.90/91 in the match-up condition of ±2°, it is smaller in the stricter condition, ±0.1°. Regarding the spatial variability in $CH_4$, the results obtained with the stricter match-up condition are more reliable, and V03.00 improves the quality of $XCH_4$. The standard deviations of the $XCH_4$ differences between GOSAT and TCCON are similar for V02.90/91 and V03.00. Considering these validation results and the improvement in fitting accuracies, the quality of the $XCH_4$ from V03.00 is comparable to or better than that from V02.90/91. In addition, the investigation of site-to-site and temporal variability of $XCO_2$ and $XCH_4$ biases from V03.00 demonstrates that their site-to-site variabilities are the approximately same level as, and the temporal variabilities are slightly smaller than those from V02.90/91.

The long-term trends of $XCO_2$ from both product versions are evaluated via in situ measurements. The V03.00 product resolves the issue that the decadal $CO_2$ growth rate estimated from the V02.90/91 products over the ocean is 1.7 ppm/decade lower than that from the in situ measurements.

Although the V03 retrieval algorithm has an issue to be resolved for $XCO_2$ over the ocean, the objectives of the update, increase in data, and improvement of the fitting accuracy are generally achieved over land. Notably, the increase in data of 13% over land and the improvements of the temporal variabilities of biases are helpful for the flux inversions or emission estimates of $CO_2$ and $CH_4$. NIES plans to release the L2 V03.00 product and the bias-corrected V03.05 in near future.

## Appendix A. Sensitivity analysis for updated items

Although the updated items do not independently affect the retrieval results of V03 and it is difficult to evaluate separately, we performed some sensitivity test retrievals in order to investigate the changes in retrieval results from each updated item as a reference. Updated items are categorized as follows:

1. cirrus properties are added to the state vector instead of the cirrus cloud screening using the 2 μm band
2. degradation model is replaced
3. solar irradiance spectra are replaced
4. gas absorption coefficient tables and the empirical noise model are replaced

The sensitivity test retrievals were performed by changing the items updated from V02.90/91 for six patterns (A – E) as listed in Table A1. The thresholds of the spectral residuals in the post-screening are same as that of V03. Since all the items are updated, A is equal to the V03.00 product. 20% of all the data until May 2020 are processed because of the computational costs. The results of the retrievals are summarized in Table A2.

The retrieval pattern A and B show similar spectral residuals (not shown) and the cirrus cloud treatment seems to have fewer impacts on the residuals. AOT of a large particle at 1.6 μm over land and temperature shift in high altitudes estimated from A are smaller than those from B. The retrieval tests C, D, and E are compared with B to estimate the effects of items 2, 3, and 4 in the following.

C is the retrieval test changing the degradation model to the old one from B. In this retrieval, few products are available over the ocean because the normalized mean squared residuals in the $O_2$ A and $SCO_2$ sub-bands are > 2, and data is rejected

in the post-screening in most observations. The residuals in the $WCO_2$ sub-band are also larger than those from B. These are because there are large differences in the baselines of the previous and updated solar irradiance spectra. The degradation models include the absolute degradation factor which adjusts the baseline of the calculated and observed spectra. Although the absolute degradation factor is estimated using the updated solar irradiance in the updated degradation model, that is done using the old solar irradiance spectra in the old degradation model. It is estimated that the residuals in the $O_2$ A sub-band are increased along with the change of the retrieved $XCO_2$ due to the residuals in the $SCO_2$ sub-band.

The number of data over the ocean from D is decreased because of the same reason as C. $XCH_4$ from D is approximately 7 ppb larger than that from B over land. Over the ocean, $XCO_2$ significantly changed. Since here are no significant differences in the number of data in time over the ocean, this implies that the update of solar irradiance has significant impacts on $XCO_2$. AOT over the ocean from D is larger than that from B.

The changes in temperature shift are largest in E. Especially, that is 1 K smaller than that from B over the ocean and the averaged values are negative globally similar to those from V02.90/91. Although there is a difference in $XCH_4$ between E and B over the ocean, that is not seen over land. Therefore, the update of the absorption coefficient may have a less direct impact on $XCH_4$.

Based on the retrieval results, the update of solar irradiance spectra seems to have a relatively large impact on the $XCH_4$ because of the significant change over land. The update of the degradation model also has the impacts on $\Delta Ps$ same as solar irradiance, but less impact on $XCH_4$. Large particle AOT is mainly affected by the updates in the treatment of cirrus clouds over land.

Over the ocean, it is very difficult to estimate the causes of the changes in the results because the numbers of data are significantly different for each pattern. However, $XCO_2$ is significantly changed by the update of solar irradiance. AOT estimated from C, D, and E are changed from that from B, so that the changes of AOT are multiply affected by the updates. Temperature shift seems to be largely affected by the updates of the absorption coefficient table over both surfaces.

## Appendix B. Information of TCCON and in situ measurement data

Site information of each TCCON data used for validation and bias correction is listed in Table B1 and B3. Data availabilities and citations of in situ measurements are listed in Table B2.

**Acknowledgments**

We acknowledge TCCON, CONTRAIL, and NOAA Global Monitoring Network for making the data available to the public. The TCCON data were obtained from the TCCON Data Archive hosted by CaltechDATA at https://tccondata.org. Data availabilities of the in situ measurement data are listed in Table B2.

**Author contributions**

YS performed the investigation of the retrieval results, contributed to the development of the retrieval system, and prepared the manuscript. YY designed and developed the retrieval system, contributed to the investigation of the retrieval results, and edited the manuscript. HO and IM contributed to the development of the retrieval system and provided the TCCON data. SN, AK, and HM contributed to the investigation of the retrieval results. TM acquired funding. JL, VV, BH, YT, MS, RK, MZ, YO, ND, and DG contributed to provide the TCCON data. All the co-authors reviewed the manuscript.

**Financial support**

This work is supported by the NIES GOSAT project. A portion TCCON data development was carried out at the Jet Propulsion Laboratory, California Institute of Technology, under a contract with the National Aeronautics and Space Administration (80NM0018D0004). The TCCON stations at Rikubetsu, Tsukuba, and Burgos are supported in part by the GOSAT series project. Local support for Burgos is provided by the Energy Development Corporation (EDC, Philippines). The Paris TCCON site has received funding from Sorbonne Université, the French research center CNRS, the French space agency CNES, and Région Île-de-France. The TCCON site at Réunion Island has been operated by the Royal Belgian Institute for Space Aeronomy with financial support since 2014 by the EU project ICOS-Inwire and the ministerial decree for ICOS (FR/35/IC1 to FR/35/C6) and local activities supported by LACy/UMR8105 and by OSU-R/UMS3365 – Université de La Réunion". The Anmyendo TCCON site is funded by the Korea Meteorological Administration Research and Development Program under grant KMA2018-00522 and KMI2022-01610.

**Conflicts of interest**

The authors have no conflicts of interest, financial or otherwise, related to this study.

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

**Table 1: Summary of the optical parameter updates**

| Gas absorption | Reference database | |
| --- | --- | --- |
| | V02 | V03 |
| $O_2$ | Tran et al. (2006) Tran and Hartmann (2008) | ABSCO V5.0 (Drouin et al., 2017) |
| $CO_2$ | Lamouroux et al. (2010) | |
| $CH_4$ | HITRAN 2008 (Rothman et al., 2009) | Devi et al. (2015; 2016) for the $2v_3$ band of $^{12}CH_4$ HITRAN 2016 (Gordon et al., 2017) for the others |
| $H_2O$ | HITRAN 2008 | ATM line list 2014 (Toon, 2015a) |
| $H_2O$ continuum | MT_CKD V2.5.2 (Mlawer et al., 2012) | MT_CKD V3.2 |

**Table 2: Retrieval setup for the V03 product. COT and CTP are additional parameters from V02.**

| State vector | No. of elements | a priori | Uncertainty |
| --- | --- | --- | --- |
| $CO_2$ mixing ratio | 15 | NIES-TM | estimated from NIES-TM |
| $CH_4$ mixing ratio | 15 | NIES-TM | estimated from NIES-TM |
| $H_2O$ mixing ratio | 15 | GPV (GSM) | estimated from GPV (GSM) |
| AOT (small particle) | 6 | SPRINTARS | 0.5 |
| AOT (large particle) | 6 | SPRINTARS | 0.5 |
| COT | 1 | 0.1 | 0.05 |
| CTP | 1 | 150 hPa | 30 hPa |
| Surface pressure | 1 | GPV (GSM) | 5 hPa |
| Temperature shift | 1 | 0 K | 5 K |
| Surface albedo (over land) | 2, 9, 11, 2, 2 ($O_2$ A, $WCO_2$, $CH_4$, $H_2O$, $SCO_2$ sub-band) | estimated from measured spectra | 1 |
| Wind speed (over ocean) | 1 | GPV (GSM) | estimated from GPV (GSM) |
| Zero level offset | 1 ($O_2$ A sub-band only) | 0 W/cm$^2$/sr/cm$^{-1}$ | $10^{-8}$ W/cm$^2$/sr/cm$^{-1}$ |
| Wavenumber dispersion factor | 4 ($O_2$ A, $WCO_2$, $CH_4$, $SCO_2$ sub-band) | 0 | $10^{-5}$ |

**Table 3: Summary of the pre/post-screening procedures for the V02 and V03 algorithm. The observation is rejected if more than one item satisfies the criteria.**

| | Item | Rejection criteria | |
|---|---|---|---|
| | | V02 | V03 |
| Pre-screening | L1B quality | Bad | |
| | Out-of-band spectrum | Outlier | |
| | CAI cloud flag | Cloudy | |
| | CAI coherent (ocean) | Cloudy | |
| | 2 μm band cloud flag | Cloudy | – |
| | Solar zenith angle | > 70° | |
| | SNR | < 70 for $O_2A$ sub-band | |
| | Land fraction | 0 % < and < 60 % | |
| Post-screening | No. of iteration | 20 | |
| | Mean squared residuals ($O_2$ A, WCO$_2$, CH$_4$, and SCO$_2$ sub-bands) | > 1.2, 1.2, 1.3, and 1.4 | > 1.2, 1.2, 1.2, and 1.3 |
| | Degree of freedom for signal | < 1 | |
| | AOT (1.6 μm) | > 0.1 | |
| | Blended albedo | > 1 | |
| | Surface wind speed | < 0.1 m/s or > 20 m/s | |
| | Absolute difference between retrieved and a priori Ps | > 20 hPa | |
| | retrieved COT | – | > 0.1 |

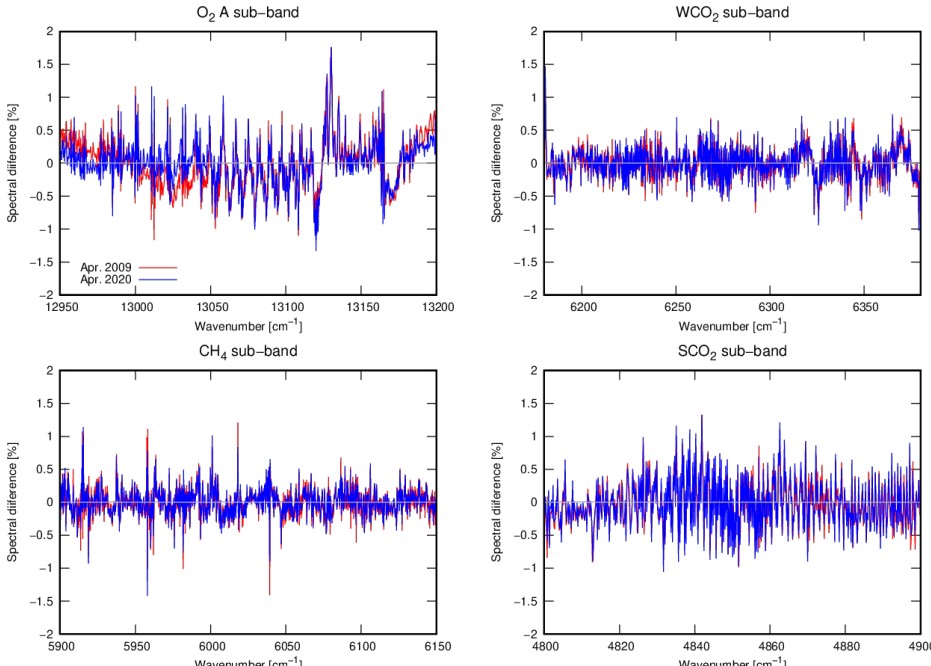

**Fig. 1: Averaged spectral residuals (simulated minus observed) normalized with the maximum radiance in each range at O₂ A (top-left), WCO₂ (top-right), CH₄ (bottom-left), and SCO₂ (bottom-right) sub-bands in April 2009 (red) and April 2020 (blue) over land obtained from V02.90.**

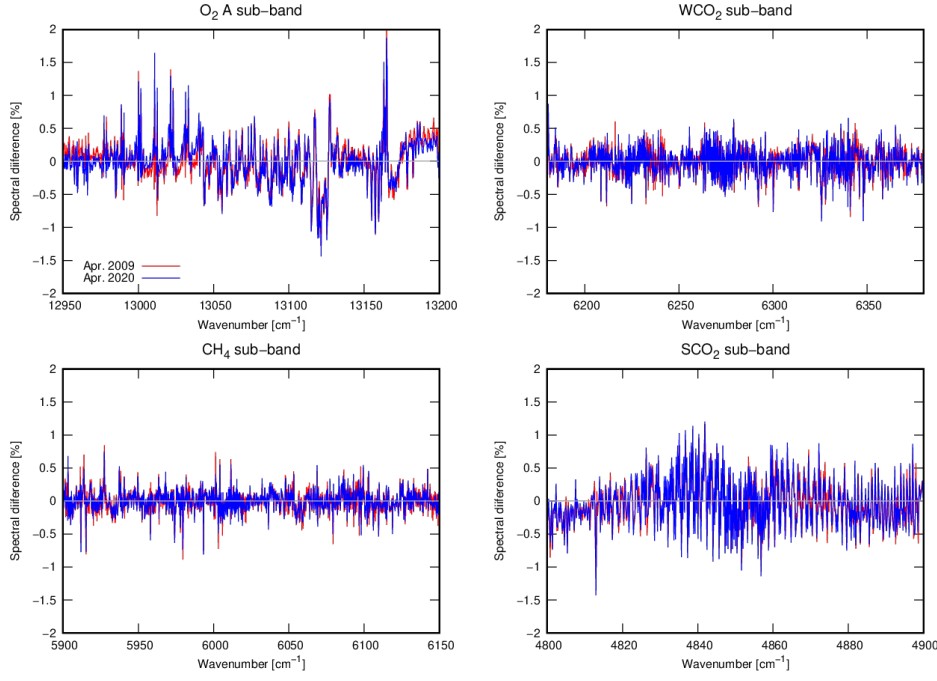

**Fig. 2: Same as Fig. 1 but for V03.00.**

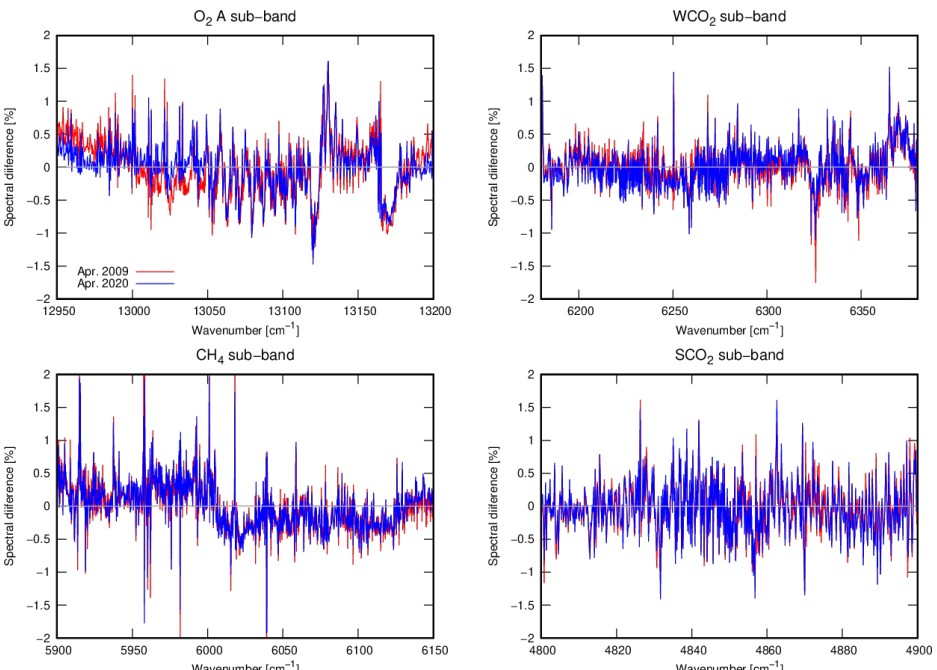

**Fig. 3: Same as Fig. 1 but for over the ocean.**

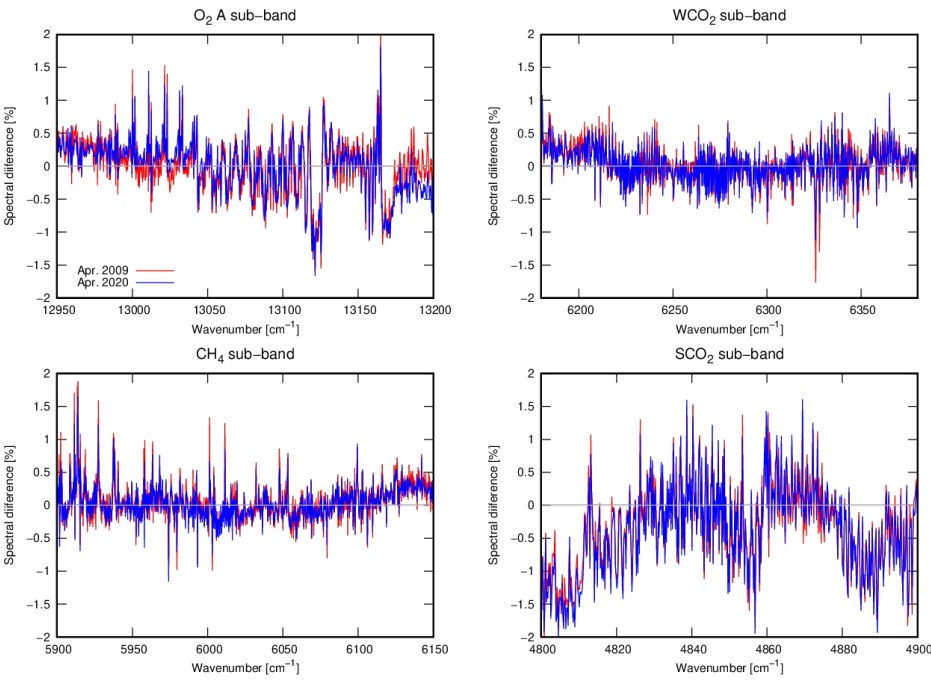

**Fig. 4: Same as Fig. 3 but for V03.00.**

**Table 4: Root mean squares of the averaged spectral residuals for each sub-band in April 2020. The unit is ×10⁻⁹ W/cm²/sr/cm⁻¹.**

|  |  | $O_2$ A | $WCO_2$ | $CH_4$ | $SCO_2$ |
|---|---|---|---|---|---|
| V02.90/91 | Land | 1.429 | 0.973 | 0.811 | 0.834 |
|  | Ocean | 1.219 | 0.859 | 0.980 | 0.762 |
| V03.00 | Land | 1.217 | 0.854 | 0.612 | 0.799 |
|  | Ocean | 1.161 | 0.687 | 0.613 | 1.206 |

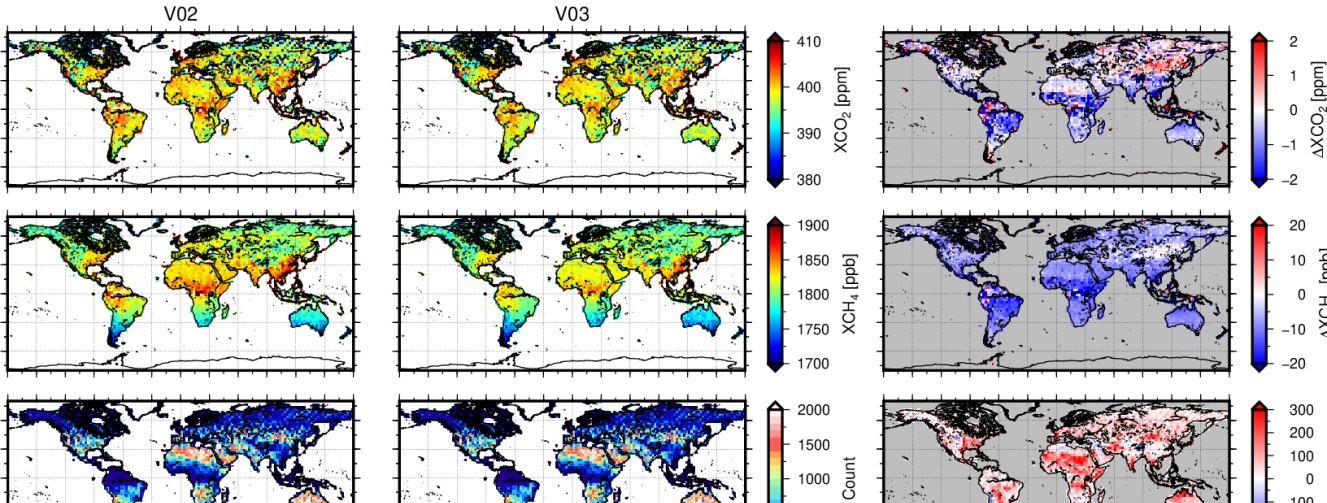

**Fig. 5: Global distributions of XCO₂ (top), XCH₄ (middle), and the number of observations (bottom) for V02.90/91 (left), V03.00 (middle), and their differences (right) from the launch to 2021 over land and mixed surface. The values are averaged or integrated within 2.5°×2.5° grid boxes. All the observations were used for each version.**

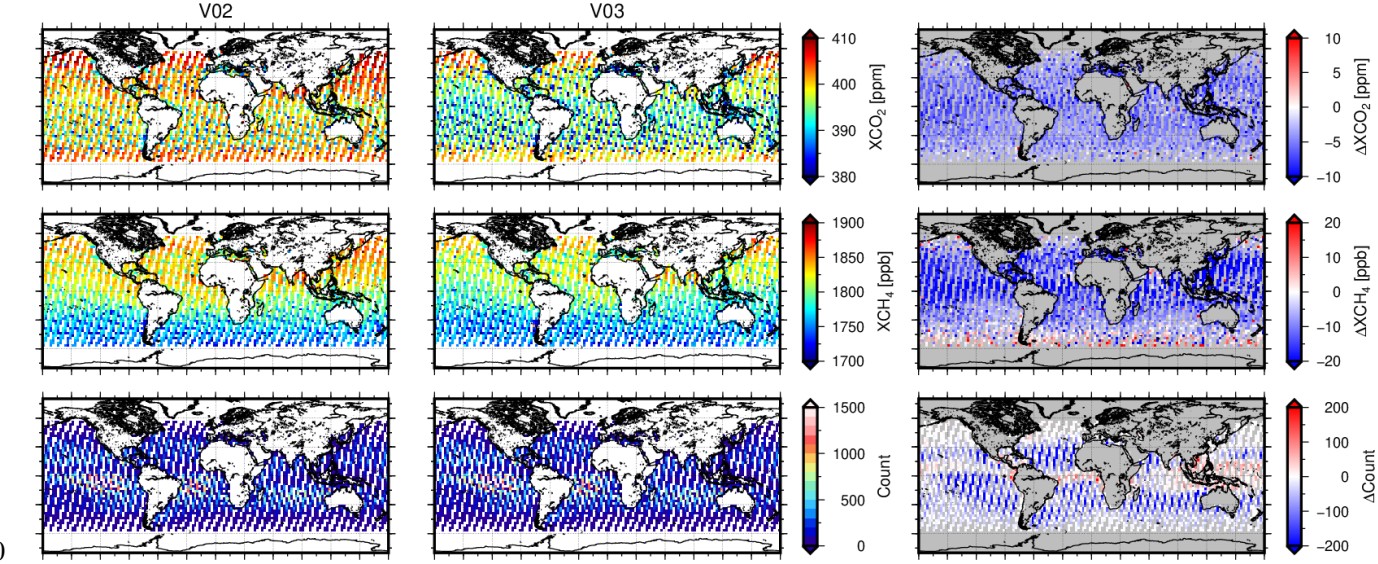

**Fig. 6: Same as Fig. 5 but for over the ocean.**

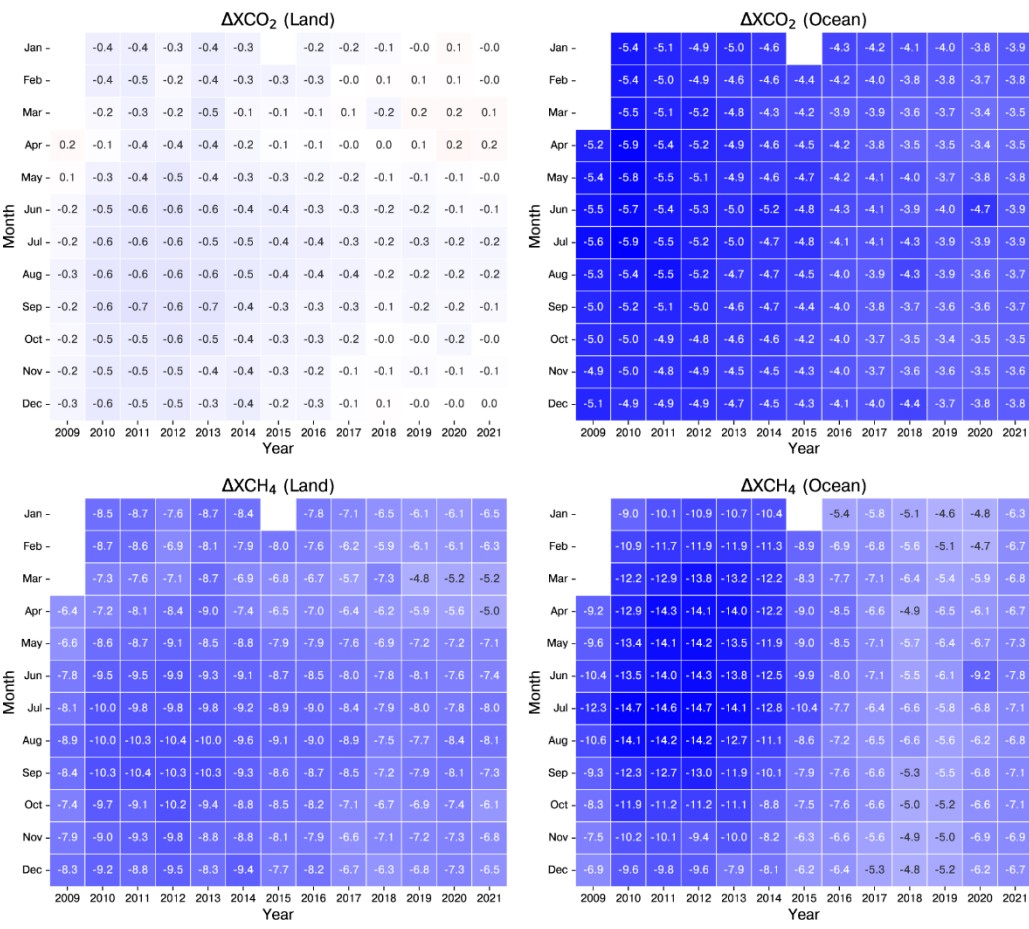

**Fig. 7: Time series heatmaps of the differences in monthly mean XCO₂ and XCH₄ over land and the ocean between V02.90/91 and V03.00 (V03.00 minus V02.90/91). Only the observations commonly available for both versions were used. The units for XCO₂ and XCH₄ are ppm and ppb.**

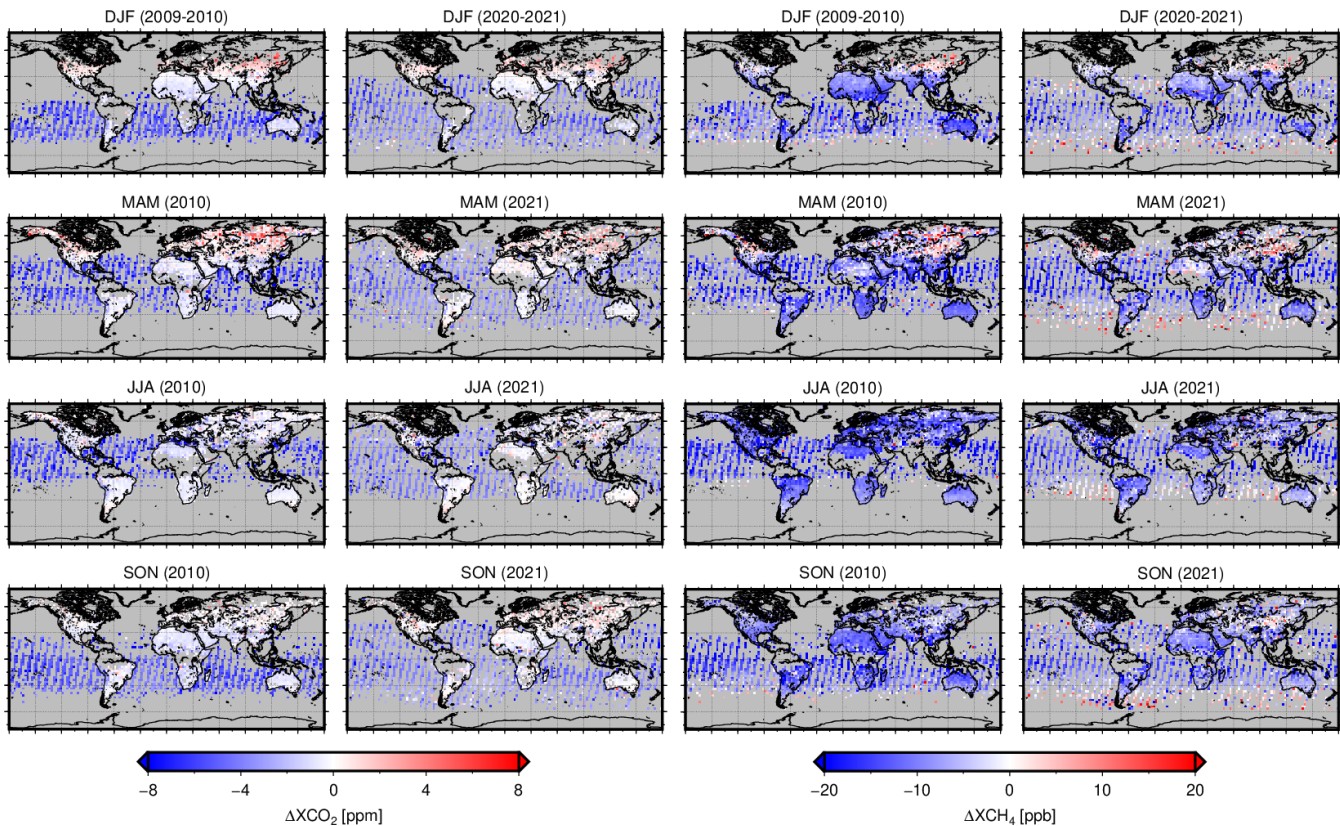

**Fig. 8 Global distributions of the differences in seasonal mean XCO₂ (left eight panels) and XCH₄ (right eight panels) in 2010 and 2021. The grid size is same as Fig. 5. All the observations were used for each version.**

**Table 5: Number of observations from the V02.90/91 and V03.00 XCO₂ products and their differences for each surface type from the launch to 2021.**

|  | V02.90/91 | V03.00 | Difference (%) |
|---|---|---|---|
| Land | 960,394 | 1,082,768 | +12.7% |
| Ocean | 557,488 | 444,477 | −20.3% |
| Mixed | 130,836 | 159,960 | +22.3% |
| Total | 1,648,718 | 1,687,205 | +2.3% |

775

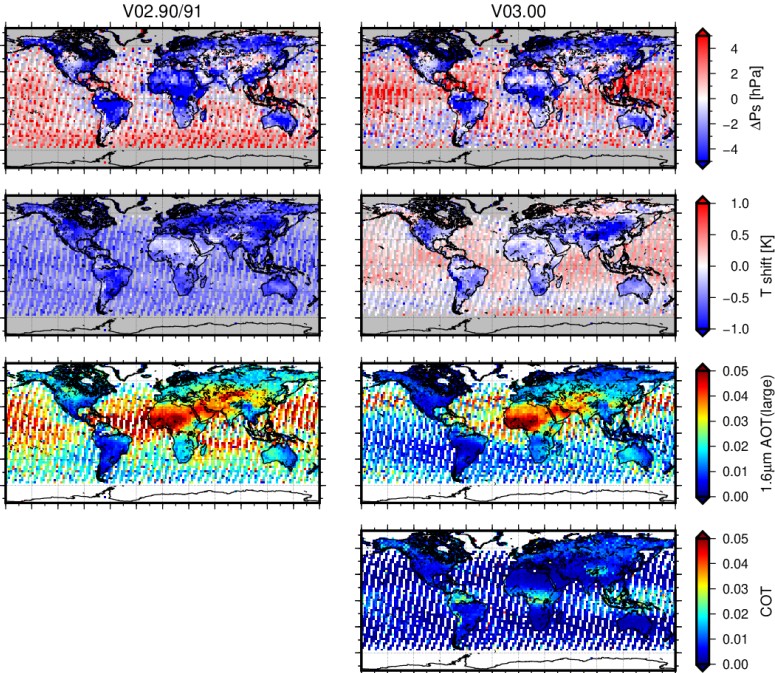

Fig. 9: Distributions of the averaged ΔPs, T shift, large particle AOT at 1.6 μm, and COT at 0.55 μm from V02.90/91 and V03.00. COT is obtainable only from V03.

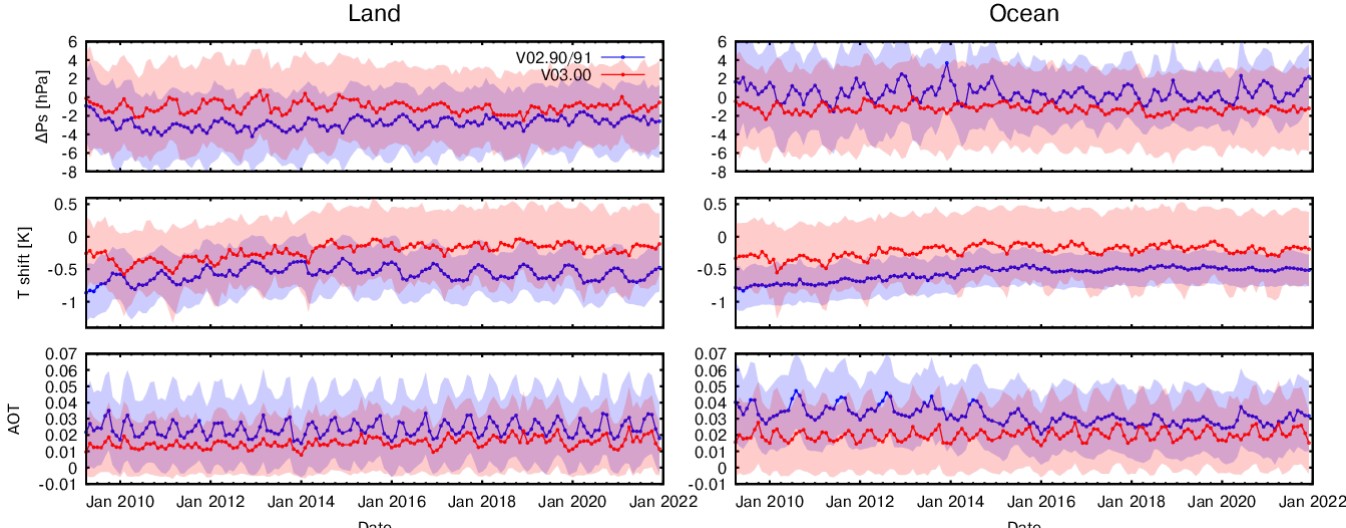

Land           Ocean

**Fig. 10: Time series of the monthly averaged ΔPs, T shift, large particle AOT at 1.6 μm from V02.90/91 and V03.00. The shades show ±1σ.**

**Table 6: Validation results of V03.00 and V02.90/91 against the TCCON measurements version GGG2020 with the match-up condition of ±2°. The mean values of the differences between TCCON and GOSAT (Bias) and their standard deviations (SD) are shown for each combination of surface conditions and gain settings.**

| Version | Surface/Gain | CO₂ | | | CH₄ | | |
|---|---|---|---|---|---|---|---|
| | | No. of data | Bias (ppm) | SD (ppm) | No. of data | Bias (ppb) | SD (ppb) |
| V02.90/91 | Land/H | 7357 | -0.56 | 2.13 | 7365 | 2.97 | 11.94 |
| | Land/M | 1385 | -0.79 | 1.89 | 1385 | 8.13 | 19.17 |
| | Ocean/H | 72 | -1.63 | 2.62 | 72 | 5.60 | 15.43 |
| V03.00 | Land/H | 8780 | -0.61 | 2.20 | 8790 | -4.23 | 11.97 |
| | Land/M | 1360 | -0.88 | 1.97 | 1360 | -0.19 | 19.29 |
| | Ocean/H | 61 | -8.12 | 2.81 | 61 | -9.71 | 14.60 |

**Table 7: Validation results of V03.00 and V02.90/91 over land with gain H against the TCCON measurement version GGG2020 in the match-up condition of ±0.1°.**

| Version | CO₂ | | | CH₄ | | |
|---|---|---|---|---|---|---|
| | No. of data | Bias (ppm) | SD (ppm) | No. of data | Bias (ppb) | SD (ppb) |
| V02.90/91 | 1743 | -0.31 | 1.76 | 1744 | 4.81 | 9.81 |
| V03.00 | 2111 | -0.43 | 1.81 | 2112 | -3.30 | 9.68 |

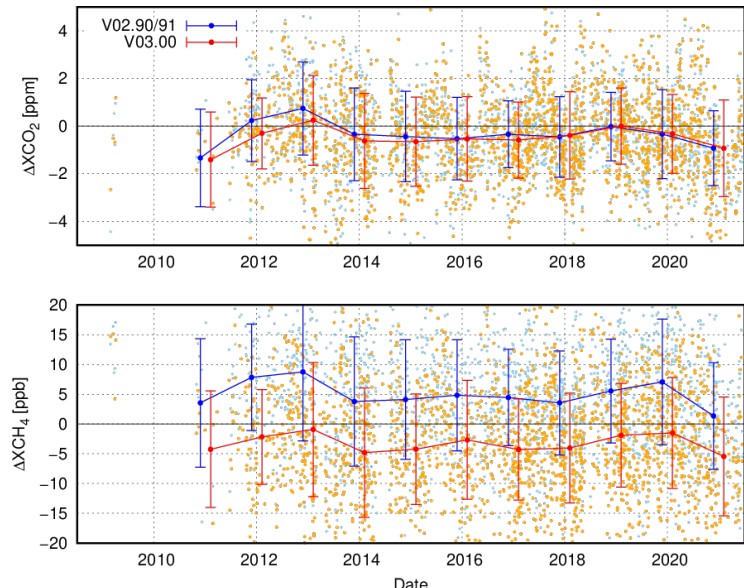

**Fig. 11: Annual mean differences of the GOSAT L2 product and TCCON GGG2020 in the match-up condition of ±0.1°. for XCO₂ (top) and XCH₄ (bottom). The red and blue lines indicate V03.00 and V02.90/91, respectively. The annual mean plots are slightly shifted between V02 and V03 for visibility. Each individual observation from V03.00 and V02.90/91 is plotted as orange and light-blue dots.**

795

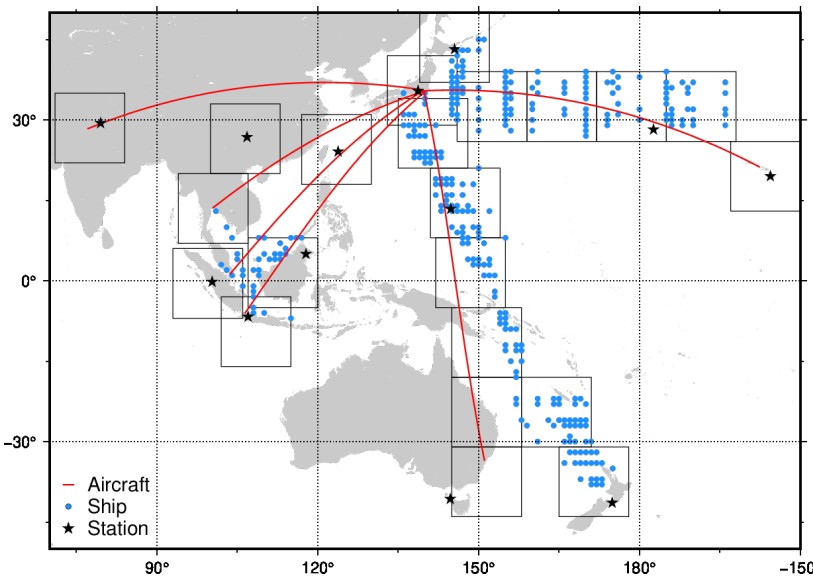

**Fig. 12: Main route or locations of the aircraft (red line), ship (blue dot), and station (black star) measurements. The areas for comparison with the GOSAT data are shown in boxes. Here, we only show the main routes of the aircraft measurements (Haneda/Narita to Delhi, Bangkok, Singapore, Jakarta, Sydney, and Honolulu) accounting for more than 97% although the remaining data contain the other routes (Haneda/Narita to Taipei, Kuala Lumpur, Denpasar, Cairns, Brisbane, and Guam).**

800

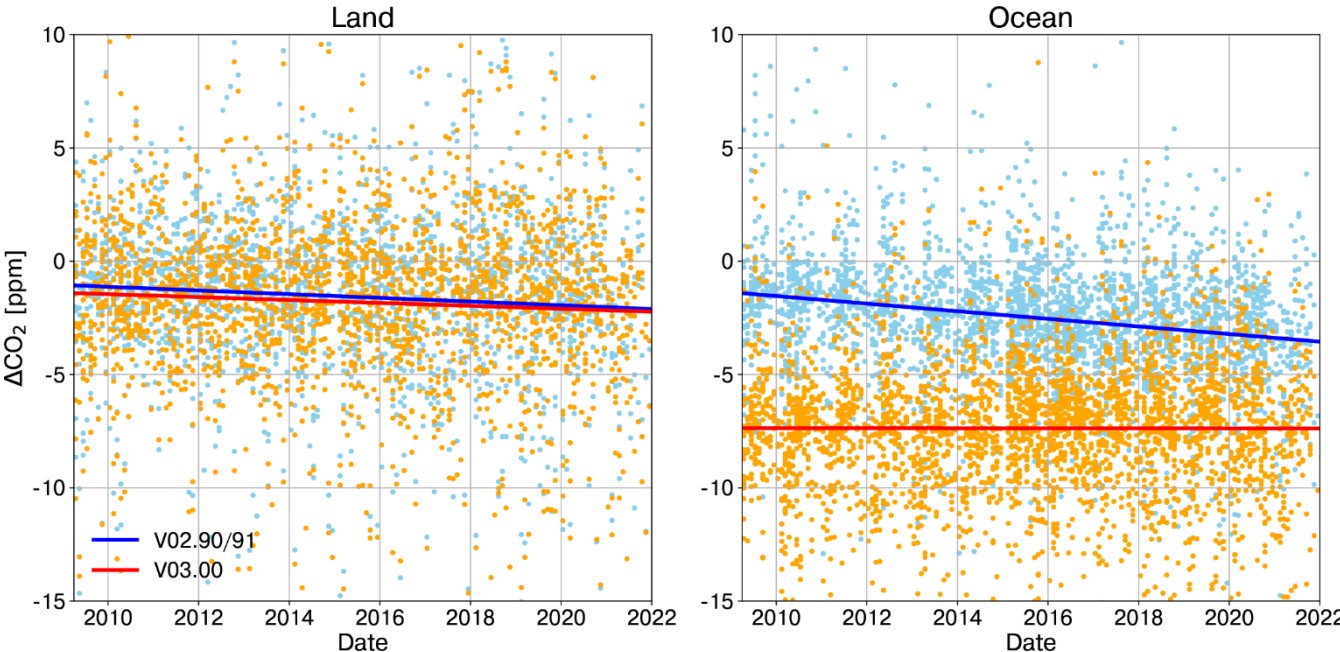

**Fig. 13: Time series of the differences between the GOSAT products and in situ measurements (GOSAT minus in situ measurements) over land (left panels) and the ocean (right panels). The regression lines are plotted as red and blue lines.**

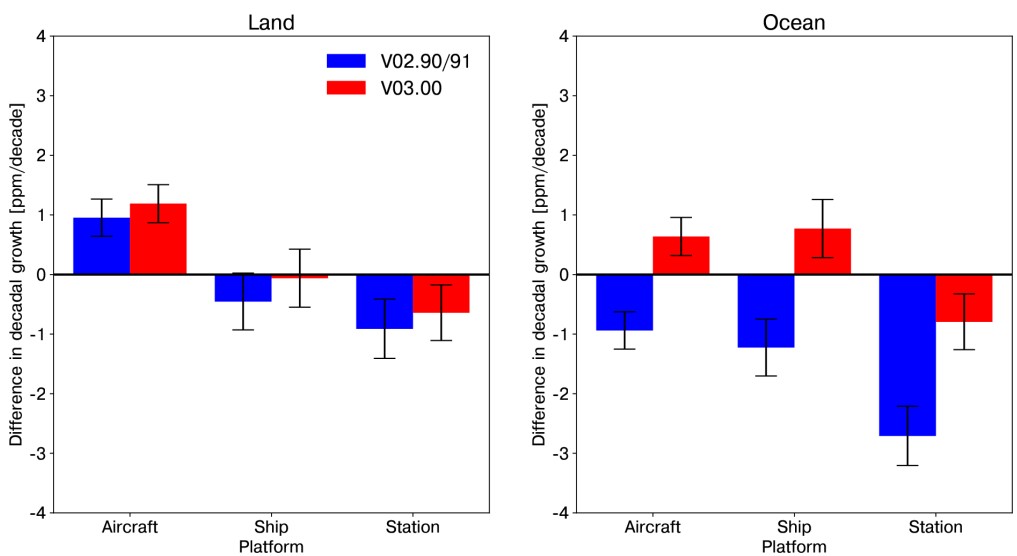

**Fig. 14: Differences in the decadal growths of CO₂ between the GOSAT product and each in situ measurement platform.**

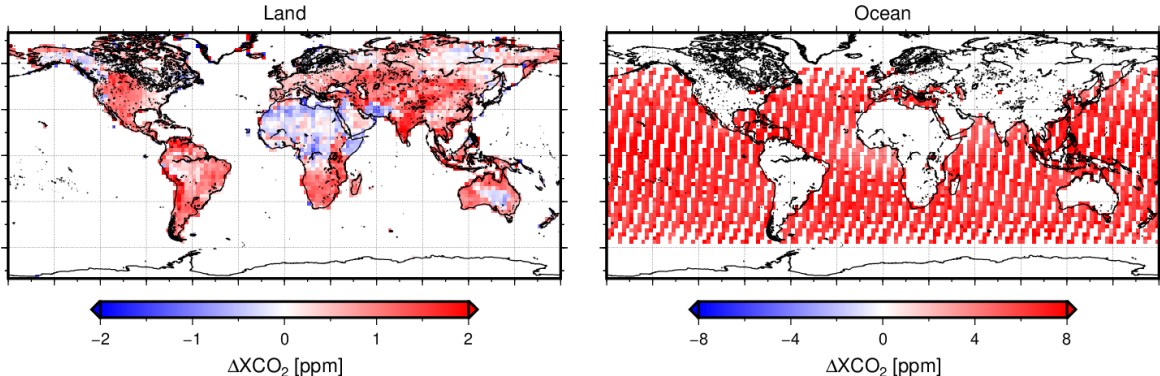

**Fig. 15: Difference of XCO₂ between V03.00 and bias-corrected one (bias-corrected minus V03.00) over land (left) and the ocean (right) averaged from the launch to 2021 within 2.5°×2.5° grid boxes.**

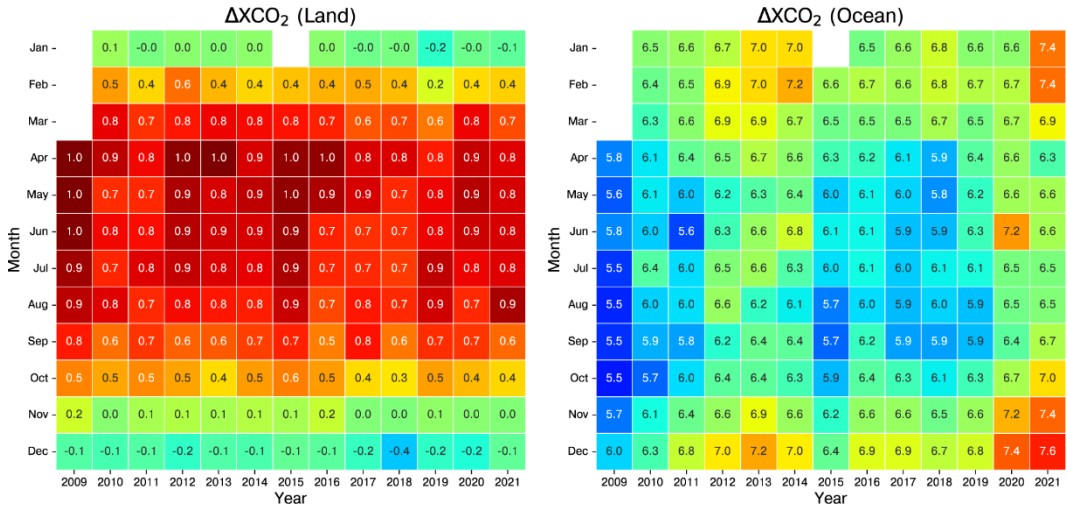

**Fig. 16 Time series heatmap of the monthly mean changes of XCO₂ by the bias correction over land (left) and the ocean (right). The color scales differ for the panels.**

**Table A1: Applied updated items (1 – 4) for sensitivity test retrievals (A – E).**

|   | 1 | 2 | 3 | 4 |
|---|---|---|---|---|
| A | x | x | x | x |
| B |   | x | x | x |
| C |   |   | x | x |
| D |   | x |   | x |
| E |   | x | x |   |

**Table A2: Summary of sensitivity test retrieval results**

|       |   | No. of data | $XCO_2$ [ppm] | $XCH_4$ [ppb] | $\Delta Ps$ [hPa] | T shift [K] | AOT (large) |
|-------|---|-------------|---------------|---------------|-------------------|-------------|-------------|
| Land  | A | 188683      | 396.79        | 1797.67       | -1.95             | -0.37       | 0.020       |
|       | B | 171359      | 396.65        | 1795.96       | -2.06             | -0.24       | 0.024       |
|       | C | 170224      | 397.31        | 1798.77       | -3.70             | -0.23       | 0.024       |
|       | D | 170393      | 396.33        | 1802.86       | -0.73             | -0.26       | 0.024       |
|       | E | 158166      | 396.56        | 1794.53       | -2.33             | -0.74       | 0.023       |
| Ocean | A | 75165       | 393.87        | 1790.36       | 1.34              | 0.18        | 0.019       |
|       | B | 56869       | 393.53        | 1788.49       | 0.86              | 0.17        | 0.015       |
|       | C | 14          | 393.19        | 1795.57       | -5.15             | -0.09       | 0.026       |
|       | D | 34059       | 401.80        | 1805.63       | 2.53              | -0.19       | 0.024       |
|       | E | 19847       | 393.26        | 1802.44       | 0.89              | -0.75       | 0.027       |

820

**Table B1: Site information of the TCCON GGG2020 data used for validation.**

| Site | Latitude | Longitude | Start date | End date | Reference |
|---|---|---|---|---|---|
| Bremen | 53.10N | 8.85E | 1/6/2009 | 6/24/2021 | Notholt et al. (2022) |
| Burgos | 18.533N | 120.650E | 3/3/2007 | 4/30/2020 | Morino et al. (2022a) |
| Caltech (Pasadena) | 34.136N | 118.127W | 9/20/2012 | 3/1/2022 | Wennberg et al. (2022a) |
| East Trout Lake | 54.354N | 104.987W | 10/3/2016 | 3/6/2022 | Wunch et al. (2022) |
| Four Corners | 36.707N | 108.48W | 3/16/2013 | 10/3/2013 | Dubey et al. (2022a) |
| Indianapolis | 39.861N | 86.004W | 8/23/2012 | 12/1/2012 | Iraci et al. (2022) |
| JPL02 | 34.202N | 118.175W | 5/19/2011 | 5/14/2018 | Wennberg et al. (2022b) |
| Karlsruhe | 49.100N | 8.439E | 1/15/2014 | 12/22/2021 | Hase et al. (2022) |
| Lauder01 | 45.038S | 169.684E | 6/28/2004 | 2/19/2010 | Sherlock et al. (2022a) |
| Lauder02 | 45.038S | 169.684E | 1/2/2013 | 9/30/2018 | Sherlock et al. (2022b) |
| Lauder03 | 45.038S | 169.684E | 10/2/2018 | 3/30/2021 | Pollard et al. (2022) |
| Lamont | 36.604N | 97.486W | 7/6/2008 | 2/27/2022 | Wennberg et al. (2022c) |
| Manaus | 3.213S | 60.598W | 9/30/2014 | 7/27/2015 | Dubey et al. (2022b) |
| Nicosia | 35.141N | 33.381E | 9/3/2019 | 6/1/2021 | Petri et al. (2022) |
| Orleans | 47.97N | 2.113E | 8/29/2009 | 3/8/2021 | Warneke et al. (2022) |
| Paris | 48.846N | 2.356E | 9/23/2014 | 6/16/2021 | Té et al. (2022) |
| Park Falls | 45.945N | 90.273W | 5/26/2004 | 2/28/2022 | Wennberg et al. (2022d) |
| Reunion | 20.901S | 55.485E | 3/1/2015 | 7/18/2020 | De Mazière et al. (2022) |
| Rikubetsu | 43.457N | 143.766E | 6/24/2014 | 6/30/2021 | Morino et al. (2022b) |
| Saga | 33.241N | 130.288E | 7/28/2011 | 6/30/2021 | Shiomi et al. (2022) |
| Sodankyla | 67.367N | 26.631E | 3/5/2018 | 10/18/2021 | Kivi et al. (2022) |
| Tsukuba | 36.051N | 140.122E | 3/28/2014 | 6/28/2021 | Morino et al. (2022c) |
| Xianghe | 39.75N | 116.96E | 6/14/2018 | 11/30/2021 | Zhou et al. (2022) |

**Table B2: In situ measurement data availability.**

| Platform/site | | Citation |
|---|---|---|
| Aircraft | | Atmospheric CO2 mole fraction data of CONTRAIL-CME, DOI: 10.17595/20180208.001 |
| Ship | | https://soop.jp (partially on request) |
| NIES station | | |
| | Ochi-ishi | Continuous Observational Data of Atmospheric CO2 Mixing Ratios at Cape Ochi-ishi, DOI: 10.17595/20160901.002 |
| | Mt. Fuji | Daily Observational Data of Atmospheric CO2 Mixing Ratios at the Summit of Mt. Fuji, DOI: 10.17595/20170616.001 |
| | Nainital | Atmospheric Carbon Dioxide Dry Air Mole Fraction at Nainital, India, DOI: 10.17595/20220301.001 |
| | Hateruma | Continuous Observational Data of Atmospheric CO2 Mixing Ratios on Hateruma Island, DOI: 10.17595/20160901.001 |
| | Guiyang | On request |
| | Danum Valley | On request |
| | Bukit Kototabang | On request |
| | Serpong | On request |
| | Bogor | On request |
| | Cibeureum | On request |
| NOAA flask/in-situ | | Atmospheric Carbon Dioxide Dry Air Mole Fractions from the NOAA GML Carbon Cycle Cooperative Global Air Sampling Network, 1968-2021, Version: 2022-07-28, DOI: 10.15138/wkgj-f215 |
| | Midway | Atmospheric Carbon Dioxide Dry Air Mole Fractions at Sand Island, Midway, https://gml.noaa.gov/aftp/data/trace_gases/co2/flask/surface/txt/co2_mid_surface-flask_1_ccgg_month.txt |
| | Guam | Atmospheric Carbon Dioxide Dry Air Mole Fractions at Mariana Islands, Guam, https://gml.noaa.gov/aftp/data/trace_gases/co2/flask/surface/txt/co2_gmi_surface-flask_1_ccgg_month.txt |
| | Cape Grim | Atmospheric Carbon Dioxide Dry Air Mole Fractions at Cape Grim, Tasmania, https://gml.noaa.gov/aftp/data/trace_gases/co2/flask/surface/txt/co2_cgo_surface-flask_1_ccgg_month.txt |
| | Baring Head | Atmospheric Carbon Dioxide Dry Air Mole Fractions at Baring Head, New Zealand, https://gml.noaa.gov/aftp/data/trace_gases/co2/flask/surface/txt/co2_bhd_surface-flask_1_ccgg_month.txt |

825

**Table B3: Site information of the TCCON GGG2014 data used for bias correction.**

| Site | Latitude | Longitude | Start date | End date | Reference |
|---|---|---|---|---|---|
| Ascension | 7.916S | 14.333W | 5/22/2012 | 10/31/2018 | Feist et al. (2014) |
| Anmeyondo | 36.538N | 126.331E | 2/2/2015 | 4/18/2018 | Goo et al. (2014) |
| Bialystok | 53.23N | 23.025E | 3/1/2009 | 10/1/2018 | Deutscher et al. (2015) |
| Bremen | 53.10N | 8.85E | 1/22/2010 | 2/24/2021 | Notholt et al. (2014) |
| Burgos | 18.533N | 120.650E | 3/3/2017 | 3/31/2020 | Morino et al. (2018a) |
| Caltech (Pasadena) | 34.136N | 118.127W | 9/20/2012 | 12/29/2020 | Wennberg et al. (2015) |
| Darwin | 12.425S | 130.892E | 8/28/2005 | 4/30/2020 | Griffith et al. (2014a) |
| East Trout Lake | 54.354N | 104.987W | 10/7/2016 | 9/6/2020 | Wunch et al. (2017) |
| Edwards | 34.958N | 117.882W | 7/20/2013 | 12/31/2020 | Iraci et al. (2016a) |
| Eureka | 80.05N | 86.42W | 7/24/2010 | 7/6/2020 | Strong et al. (2017) |
| Four Corners | 36.707N | 108.480W | 3/16/2013 | 10/4/2013 | Dubey et al. (2014a) |
| Garmisch | 47.476N | 11.063E | 7/16/2007 | 4/1/2021 | Sussmann and Rettinger (2015) |
| Hefei | 31.91N | 117.17E | 9/18/2015 | 10/23/2018 | Liu et al. (2018) |
| Indianapolis | 39.861N | 86.004W | 8/23/2012 | 12/1/2012 | Iraci et al. (2016b) |
| JPL02 | 34.202N | 118.175W | 5/19/2011 | 5/14/2018 | Wennberg et al. (2016a) |
| Karlsruhe | 49.100N | 8.439E | 4/19/2010 | 12/22/2021 | Hase et al. (2015) |
| Lauder01 | 45.038S | 169.684E | 6/29/2004 | 12/9/2010 | Sherlock et al. (2014a) |
| Lauder02 | 45.038S | 169.684E | 2/2/2010 | 10/31/2018 | Sherlock et al. (2014b) |
| Lauder03 | 45.038S | 169.684E | 10/3/2018 | 12/31/2020 | Pollard et al. (2019) |
| Lamont | 36.604N | 97.486W | 7/6/2008 | 12/28/2020 | Wennberg et al. (2016b) |
| Manaus | 3.213S | 60.598W | 10/1/2014 | 6/24/2015 | Dubey et al. (2014b) |
| Nicosia | 35.141N | 33.381E | 8/31/2019 | 3/9/2021 | Petri et al. (2022) |
| Orleans | 47.97N | 2.113E | 8/29/2009 | 3/8/2021 | Warneke et al. (2014) |
| Paris | 48.846N | 2.356E | 9/23/2014 | 6/22/2020 | Té et al. (2014) |
| Park Falls | 45.945N | 90.273W | 6/2/2004 | 12/29/2020 | Wennberg et al. (2017) |
| Reunion | 20.901S | 55.485E | 9/16/2011 | 7/18/2020 | De Mazière et al. (2014) |
| Rikubetsu | 43.457N | 143.766E | 11/16/2013 | 9/30/2019 | Morino et al. (2016) |
| Saga | 33.241N | 130.288E | 7/28/2011 | 12/29/2020 | Kawakami et al. (2014) |
| Sodankyla | 67.367N | 26.631E | 5/16/2009 | 10/20/2020 | Kivi et al. (2017) |

| Tsukuba | 36.051N | 140.122E | 8/4/2011 | 9/30/2019 | Morino et al. (2018b) |
|---|---|---|---|---|---|
| Wollongong | 34.406S | 150.879E | 6/26/2008 | 6/30/2020 | Griffith et al. (2014b) |
| Zugspitze | 47.42N | 10.98E | 4/24/2015 | 4/1/2021 | Sussmann and Rettinger (2018) |