# Peer review of "Update on the GOSAT TANSO-FTS SWIR Level 2 retrieval algorithm"

_Atmospheric Measurement Techniques, 2022_

## Author Comment (AC1)

Dear anonymous referee#1,

The authors appreciate your constructive comments. We would respond to the reviewer's comments as follows. The reviewer's comments are written in *Italics* and the corresponding author's response is followed. The modified or added parts in the manuscript are highlighted in red.

Best regards,

Yu Someya

*I wonder if the forward radiative transfer model has been updated since Yoshida et al. (2011). If so, please provide some discussion on that aspect of the algorithm.*

There are no updates for the radiative transfer model from Yoshida et al. (2011). If you mean the radiative transfer model is replaced with that used in the GOSAT-2 retrievals, that is not done.

*Figs. 1-4: can the authors clarify if the residuals shown here were calculated before or after the post-processing quality control? Also, can the authors provide some metrics (other than visual inspection) based on which conclusions were drawn, for example, on the fitting quality for V2 and V3 in lines 160-165?*

The averaged residuals shown in the figures are calculated from the scans after post-screening. The sentence was modified as follows.

   Line 155: Figure 1 shows the averaged spectral residuals after the post-screening at each sub-band obtained in
   April 2009 and April 2020 over land from V02.90.

You may have reviewed the manuscript before addressing technical comments against the submitted version of the manuscript. The root mean squares of the averaged residuals are shown in Table 4 and that is mentioned in line 165 – 167. Here, we conclude that the RMSs are reduced except for the $SCO_2$ sub-band over the ocean.

*Can the authors provide more thorough analysis on the potential causes for the residuals in the strong CO2 band? Also, what measures are being considered to mitigate the negative bias in xCO2 over the ocean?*

The residuals in the $SCO_2$ sub-band are probably due to the inconsistency of baseline between the old and new solar irradiance spectra. The figure of solar irradiance spectra is added to supplementals and the summary of the sensitivity test retrievals are added as Appendix A. The following statement is added to the body text.

   Line 182: This is introduced by the update of the solar irradiance spectra because there are some inconsistencies
   in the spectral baseline between the old and updated spectra particularly in Band 3 (see Appendix A and
   supplementals).

   Line 187: Therefore, we need to precisely evaluate the calibration data such as obtained in the Railroad valley
   campaign using the updated solar irradiance spectra to improve the fitting accuracy especially on the $SCO_2$ sub-
   band.

*Figure 6: can the authors elaborate on what may have caused the changes in these parameters, in particular, AOT? Is there a time dependence in these changes?*

The summary of the sensitivity test retrievals is added as Appendix A in order to investigate the effect of each update. The temporal trends of the monthly averages of the retrieved ancillary parameters are added as Fig. 10. Based on Appendix A and Fig.10, the following sentences are added to the body text.

Line 237: Although the updated items do not independently affect the retrieval results and it is difficult to evaluate separately, the large causes of the change in the retrieved ancillary parameters are as follows from the sensitivity test retrievals (Appendix A). Temperature shift is increased globally by the update of the gas absorption coefficient tables. Surface pressure seems to be impacted by the replacement of solar irradiance because $\Delta Ps$ was changed by this update over land. The changes in surface pressure should contain two effects. One is the direct impact of the change in spectroscopy on the $O_2$ A sub-band. The other one is the impact through the change of $XCO_2$ introduced by the inconsistency of the spectral baseline in the $SCO_2$ sub-band. The behaviors of changes in AOT differ for land and the ocean. The changes in AOT are mainly affected by the addition of cirrus properties to the state vector over land. On the other hand, those over the ocean seem to be affected multiply by the updates. Figure 10 shows the time series of the ancillary parameters. V02.90/91 has a long-term temporal dependency on the retrieved surface pressure overland, temperature shift, AOT over the ocean. The pointing system of TANSO−FTS was switched from the primary system (PM−A) to the backup system (PM−B) on January 26, 2015. The trends differ for PM−A or PM−B, and they are larger in PM−A. For V03.00, those in surface pressure and AOT almost disappeared whereas that in temperature shift remains in PM−A.

*Figure 9: can the authors provide some details on how the differences were calculated between in situ measurements and GOSAT? In particular, given that the surface and airborne in-situ measurements can be quite different, what measures were taken to ensure that a coherent time-series can be constructed?*

The trends shown in Fig.13 in the revised manuscript are calculated by the monthly averaged $CO_2$ concentrations for each area from GOSAT and the in situ measurements. One plot corresponds to the difference/month/area/platform. We did not perform the specific operations to consider the differences between the platform and the averages of the observed data are directly compared. In case we investigate the spatially and temporally detailed comparisons, the analysis using the transport model should be needed. In this manuscript, we did not use the models because we estimate that the decadal growth rates for all the latitudes show similar trends even by the different platforms. The following sentence was added to the body text.

Line 320: The monthly averaged values in each area from GOSAT and the in situ measurements are directly compared to investigate the difference in the decadal growth.

*Line 282: can the authors provide some details on how the growth rate were derived from a noisy time series (and the uncertainties of the growth rate)?*

The differences in the decadal growth are simply estimated by the least-squared linear regression using the scatter plots shown in Fig. 10. The following sentence was added to the body text. In addition, the standard deviations were

also added to the estimated differences in the decadal growth.

Line 323: The trend is estimated by the least-squared linear regression from the scatter plots.

---

## Author Comment (AC2)

Dear Tommy Taylor,

The authors appreciate your constructive comments. We would respond to the reviewer's comments as follows. The reviewer's comments are written in *Italics* and the corresponding author's response is followed. The modified or added parts in the manuscript are highlighted in red.

Best regards,

Yu Someya

*Figures 1-4 (spectral residuals) need statistics. Also recommend writing out the name of each spectral band to each panel as a title or legend. Maybe adding a horizontal "zero" line would be helpful too. It could be informative to generate some "zoomed in" versions of the plots that focus on very specific spectral ranges so that the reader can get a better sense of the differences between the v2.9 and v3. Is it possible to change the vertical scale from absolute radiance units to percent of the continuum? Generally that is more intuitive and informative. Might be illustrative to list the XCO2 value from the retrievals for these specific cases in a legend. It would be somewhat interesting to see how much the XCO2 changed from v2.9 to v3.*

Fig. 1 – 4 were replotted as the normalized ones and the titles were added as the sub-band name for all the panels. The text was changed as follows.

> Line 155: Figure 1 shows the averaged spectral residuals after the post-screening at each sub-band obtained in April 2009 and April 2020 over land from V02.90. The plots were normalized with the maximum radiances in each spectral range.

Since the spectral residuals are not for the specific cases but the averaged ones, it is difficult to state the corresponding $XCO_2$ values. The changes in the $XCO_2$ from V02.90/91 to V03.00 are shown in the following sections.

*Does the NIES retrieval algorithm use EOFs to help account for mis-fitting the spectra caused by errors in absorption coefficients? If not, then perhaps it should be investigated.*

No, the NIES V03 retrieval algorithm does not use EOFs, but instead uses empirical noise. However, the retrieval code itself has the ability to retrieve EOFs simultaneously, albeit with the limitation of one EOF per sub-band. Instead of empirical noise, we attempted to retrieve EOFs simultaneously. The results showed that one EOF was not sufficient, especially for WCO2 sub-band, because the residuals were SNR-dependent. The differences in Xgas between V03.00 and EOF retrieval were small except for $XCO_2$ over the ocean. The large negative bias of V03.00 $XCO_2$ over the ocean was somewhat smaller, but still present. We will further investigate the EOF retrieval in future algorithm improvements. Thank you for the helpful comments.

*In Section 4.2, I would recommend that the authors show either a scatter plot of XCO2 from V3 vs v2.9 for matched soundings. Then apply a linear fit and report the statistics to demonstrate the difference. Alternatively, the delta*

*XCO2 between the two versions could be plotted as a time series heat map. Are there differences in time?*

Two-dimensional histograms for $XCO_2$ and $XCH_4$ are added as Fig.6. There are temporal dependencies in the difference as discussing in section 4.4. The following statements were added or modified in the body text.

Line 193: The $XCO_2$ from V03.00 over land is approximately the same as that from V02.90/91. Conversely, over the ocean, the $XCO_2$ from V03.00 is 4.24 ppm lower than that from V02.90/91 for the match-up observations.

Line 203: The temporal heatmaps of the differences in the monthly mean $XCO_2$ and $XCH_4$ between the versions are shown in Fig. 7. Differences in monthly mean $XCO_2$ get smaller with time, particularly over the ocean. This means the growth rate of $XCO_2$ from V03.00 is larger than that from V02.90/91. The long-term trend in $XCO_2$ is evaluated using the in situ measurements in section 4.4. Similar trends are also seen in $XCH_4$. Additionally, the seasonal variabilities of $XCH_4$ are larger than those of $XCO_2$, especially for the former period over the ocean. This is partly because the changes in $XCH_4$ over the ocean have latitudinal dependencies as shown in Fig. 6.

*The maps in Fig 5 would be more useful if separate maps were made for land and ocean using different color scales. Are the current plots showing individual soundings or are these gridded values? I'm pretty sure that they are individual values, which means that there is probably a lot of overplotting in these figures, making them not all that useful. An alternative idea is to only plot a single year of soundings to reduce the amount of soundings overlap. Or plot seasonal maps (DJF, MAM, JJA, SON).*

It is averaged values within 2.5°×2.5° grid boxes. The statement was added to the caption of Fig. 5. The figure was separately illustrated as Fig. 5 over land and Fig. 6 over the ocean.

*Table 5 presents the statistics of the retrieved XCO2 for v2.9 and v3 as compared to TCCON measurements with the loose collocation requirements. These results essentially suggest that v3 XCO2 for landH is slightly worse compared to TCCON than it was for v2.9. It is probably worth investigating if this is driven by some of the TCCON sites in particular. Maybe a table showing the N/bias/scatter for v2.9 and v3 for each station individually would be useful. I notice that for the tighter collocation criteria, the overall statistics are slightly improved for v3 as compared to v2.9, which is good.*

The statistics of the comparison with TCCON are shown in Table 6 (You have probably reviewed the old version of the manuscript before technical correction). The tables of the comparison with the loose collocation requirements for the individual sites are shown in supplementals because the tables are very large. According to the tables, it seems that the specific sites derive the changes of $XCO_2$ between V02.90/91 and V03.00 although there are some differences for each site. The statistics with the stricter horizontal condition for each site were added as Table S7 and S8 in the supplementals.

*I'm not really sure what to make out of the huge XCO2 bias in v3 over Ocean. It seems like some effort needs to go into the NIES L2FP to fix this by making the treatment of the ocean surface more complicated. Clearly the retrieval is unable to make a proper fit. Because of the delta XCO2 scale and the overplotting on Fig 5, it is impossible to tell what the large negative bias really looks like. Again, recommend making the maps separate for land/ocean with appropriate color scale. And probably more useful to show data for a single year/season at the beginning and end of*

*the date record, e.g., 2010 vs 2020.*

We have estimated that the inconsistency of the spectral baseline in the retrievals is mainly introduced by the update of the solar irradiance spectra. The figure of the comparisons of the old and new solar irradiance spectra was added in supplementals and the following statement was added in the main text.

Line 182: This is introduced by the update of the solar irradiance spectra because there are some inconsistencies in the spectral baseline between the old and updated spectra particularly in Band 3 (see Appendix A and supplementals).

The differences in the seasonal mean $XCO_2$ and $XCH_4$ between the versions were added as Fig.8 and the following statements were added in the body text.

Line 208: The global distributions of the seasonal mean $XCO_2$ and $XCH_4$ in 2010 and 2021 are shown in Fig. 8. As seen in Fig. 5 − 7, the differences are smaller in the recent period, and they have latitudinal dependencies. In addition, the latitudinal variations change seasonally as shown in Fig. 8. The increasing trend of $XCO_2$ in the high latitudes in Fig. 5 is introduced by the change in boreal spring (MAM) and this is not seen in boreal summer (JJA). A similar characteristic is also seen in $XCH_4$.

*Fig 7. Some fitting statistics are needed to support the discussion.*

The decadal trends of the differences in $XCO_2$ and $XCH_4$ were added as follows in the body text.

Line 301: The decadal trends of the differences from V02.9X and V03.00 are −0.63±0.15 and −0.11±0.14 ppm/decade for $XCO_2$ and −2.41±0.84 and −0.37±0.77 ppb/decade for $XCH_4$. The consistencies of the decadal trend are slightly improved in V03.00.

*Fig 8: the shading of the land is too light. Increase the contrast between land and ocean.*

Fig. 12: The color of land areas was darkened.

*Fig 9: Recommend plotting these as density heat maps, and making into 4 panels for Land/Ocean and v2.9/v3. Also needs fitting statistics to support discussion.*

We plotted the number density separately as the following figure according to your comment. However, the data is not significantly concentrated, and the change seems not to be so effective. The overplots for both surfaces are preferable because the differences in the regression lines are easily identified. Therefore, we would like to keep the figure (Fig. 13) unchanged.

The standard deviations of the fitted trends were added in the body text (Line 327 − 328).

[Figure]

*Fig 10: Recommend separating into 2 panels for land/ocean. The current delta xco2 range of +/-10 ppm makes this plot very uninformative.*

The figure was separated into two with different color scales (Fig. 15). The corresponding statements were added ad follows.

> Line 353: Over land, the corrections are generally positive although they are negative only in the high reflectance surface areas such as Sahara and Australia. The corrections over the ocean show similar positive values globally.

*Sec 4.4 Evaluating the long term trend using in situ measurements. This section needs some corresponding plots to support the discussion. There is a lot of good work here, but the authors do not show any of it. The paper is very short, so it wont hurt to add a few more interesting plots. Some plots of the calculated CO2 growth rate are definitely needed since that was one of the big motivations for the changes to the L2FP code.*

The differences in $CO_2$ growth rate between GOSAT and each in situ measurement are added as Fig.14. The corresponding statements are added as follows.

> Line 329: The differences in the growth rates between GOSAT and each platform are shown in Fig. 14. Over land, the absolute differences in the growth rates from V03.00 are smaller than those from V02.90/91 for ship and station measurements although that is slightly larger for aircraft. Over the ocean, the differences from V03.00 are smaller than those from V02.90/91 for all platforms. In particular, the large discrepancy of -2.7 ppm/decade with the station measurement in the V02.90/91 product was improved to -0.8 ppm/decade in the V03.00 product.

As you commented, the evaluations for each platform, region, and GOSAT product version had been performed. However, we think they are out of focus in this paper. Therefore, only the evaluations for the long-term and whole area are included in this manuscript.

*Sec 4.5 Bias correction. This section also needs one or two interesting plots to support the discussion.*

Time series heatmaps of the monthly mean changes by the bias correction was added as Fig.16 and the statements are added in the body text as follows.

> Line 356: The time series of the monthly mean changes by the bias correction is depicted in Fig. 16. The seasonal dependencies of the correction differ for the surface. Over land, the correction magnitude is large in boreal spring and summer. On the other hand, that is large in boreal winter over the ocean. This is because the ancillary parameters used in the bias correction are different by the surfaces and the common parameters also have different seasonal variations by the surfaces as shown in Fig. 10.

*Line 55 in Introduction: please cite [Taylor, ESSD, 2022] for the ACOS L2FP retrieval as applied to GOSAT.*

The reference was added (Line 55).

*Line 61/62. the sentence beginning with "However, the systematic structures..." should be split into 2 separate sentences for clarity.*

The statements are modified as follows.

> Line 61: However, there are still some issues to address. First, the systematic structures in the spectral residuals still exist in the retrieval results. Second, the increase of data amount in the L2 product is further required.

*Line 92: how large is the expected increase in throughput due to this change? Did the change roughly meet expectations?*

Although we can roughly estimate the increases in data after pre-screening, it is difficult to expect those after post-screening because the retrieved COT is used in the post-screening. It is not clear how the number of data increases due to the change in the cloud screening in the V03.00 product because that is affected by the other updates.

*Line 121; In the sentence "Although the new degradation model..." do you mean "Although the results from the new degradation model..."*

Here we don't mean the retrieval results but the degradation model itself (FIG. 9 and 10 in Someya et al. *JTECH,* 2020).

*Line 176: "The temporal differences are possibly due to the contributions by the other components..." What other components specifically?*

The new degradation model is constructed by principal component analysis (PCA) of the calibration spectra. The other components stated here are the components that are generated by PCA but not used in the construction of the degradation model. We couldn't determine the causes of the wavenumber-dependent components because there is not enough information (Someya et al., 2020). The statement was modified to specify the components generated by PCA.

> Line 177: The number of components of principal component analysis used to construct the degradation model in the $O_2$ A sub-band is smaller than the other band because the contributions of the primary components are large.

*Line 204; "...seems to correspond to that of the difference in XCH4 shown in Fig 5." Would be useful to generate and show a correlation plot to support this claim.*

The figure is added in supplementals. Alternatively, correlation coefficients are noted in the body text as follows.

Line 228: Correlation coefficients between the changes in the retrieved surface pressure and those in $XCH_4$ from V02.90/91 and V03.00 are −0.57 over land and −0.64 over the ocean. The relatively large decrease in $XCH_4$ in low latitudes over the ocean could be partly attributed to the changes in $\Delta Ps$. For $XCO_2$, those are −0.57 over land and −0.11 over the ocean.

*Line 229: "There are no substantial changes...although the biases are different between v3 and v2.9 in some cases". Unfortunately the results are actually slightly worse for v3.*

The statement was modified as follows.

Line 268: There are no substantial changes in the standard deviations of the differences for $XCO_2$ and $XCH_4$ in all the situations (slightly worse in V03.00), although the biases are different between V03.00 and V02.90/91 in some cases.

*Line 245/246: No statement about XCO2, which unfortunately is actually worse for v3 than v2.9 compared to TCCON.*

The following sentence was added in the body text.

Line 284: Similar to the results from the looser match-up condition, the results of $XCO_2$ from V03.00 increase the number of observations and are slightly worse biases and standard deviations.

*Line 319: "...a gap in the spectral baseline..." I dont understand this comment.*

Based on the statements in section 4.1, the sentence was modified as follows.

Line 373: This increase in the residual is mainly attributed to spectral biases at baseline between observed and simulated spectra.